# Transferring principles of solid-state and Laplace NMR to the field of *in vivo* brain MRI

João P. de Almeida Martins[1,2], Chantal M. W. Tax[3], Filip Szczepankiewicz[4,5], Derek K. Jones[3,6], Carl-Fredrik Westin[4,5], Daniel Topgaard[1,2]

[1]Division of Physical Chemistry, Department of Chemistry, Lund University, Lund, Sweden
[2]Random Walk Imaging AB, Lund, Sweden
[3]Cardiff University Brain Research Imaging Centre (CUBRIC), Cardiff University, Cardiff, United Kingdom
[4]Harvard Medical School, Boston, MA, United States
[5]Radiology, Brigham and Women's Hospital, Boston, MA, United States
[6]Mary MacKillop Institute for Health Research, Australian Catholic University, Melbourne, Australia.

*Correspondence to*: João P. de Almeida Martins (joao.martins@fkem1.lu.se)

**Abstract.** Magnetic resonance imaging (MRI) is the primary method for non-invasive investigations of the human brain in health, disease, and development, but yields data that are difficult to interpret whenever the millimeter-scale voxels contain multiple microscopic tissue environments with different chemical and structural properties. We propose a novel MRI
framework to quantify the microscopic heterogeneity of the living human brain as spatially resolved five-dimensional relaxation-diffusion distributions by augmenting a conventional diffusion-weighted imaging sequence with signal encoding principles from multidimensional solid-state nuclear magnetic resonance (NMR) spectroscopy, relaxation-diffusion correlation methods from Laplace NMR of porous media, and Monte Carlo data inversion. The high dimensionality of the distribution space allows resolution of multiple microscopic environments within each heterogeneous voxel as well as their
individual characterization with novel statistical measures that combine the chemical sensitivity of the relaxation rates with the link between microstructure and the anisotropic diffusivity of tissue water. The proposed framework is demonstrated on a healthy volunteer using both exhaustive and clinically viable acquisition protocols.

## 1 Introduction

The structure of the brain is affected by both disease and normal development over a wide range of length scales. To
measure and map the cellular architecture and molecular composition of the living human brain is a challenging experimental endeavor that promises far-reaching implications for both clinical diagnosis and our understanding of normal brain function. Over the last decades, Magnetic Resonance Imaging (MRI) methods have been crucial for the progress of neuroanatomical studies (Lerch et al., 2017). Most clinical MRI applications rely on detecting [1]H nuclei of water molecules to produce three-dimensional images with a spatial resolution on the millimeter scale. Even though the attainable resolution
is clearly insufficient for direct observation of individual cells, chemical and microstructural features can be investigated by probing their effect on magnetic resonance observables such as nuclear relaxation rates (Halle, 2006) and the translational

diffusivity (Le Bihan, 1995) of water. Relaxation and diffusion parameters can thus indirectly report on various microscopic properties, including cell density (Padhani et al., 2009), orientation of nerve fibers (Basser and Pierpaoli, 1996), and presence of nutrients (Daoust et al., 2017). Current quantitative relaxation (Tofts, 2003) and diffusion (Jones, 2010) MRI

observables are exquisitely sensitive to the cellular processes associated with knowledge acquisition (Zatorre et al., 2012), neuropsychiatric disorders (Kubicki et al., 2007), and different tumor types (Nilsson et al., 2018a), but suffer from poor specificity and the same experimental data may support several distinct biological scenarios (Zatorre et al., 2012).

More detailed information can be obtained by taking into account that each MRI voxel comprises hundreds of thousands of cells with potentially different properties, implying that the per-voxel signal may include contributions from multiple

microenvironments with distinct values of the MRI observables. To resolve the various microenvironments within a single voxel remains a highly challenging problem of vital importance for the progression of quantitative MRI studies. The signals from heterogeneous materials are often approximated as integral transformations of nonparametric distributions of relaxation rates or diffusivities (Istratov and Vyvenko, 1999), which may be estimated by Laplace inversion of data acquired as a function of the relevant experimental variable (Whittall and MacKay, 1989). Within the context of human brain MRI, the

components of the distributions have been assigned to water populations residing in specific tissue microenvironments such as myelin (Mackay et al., 1994) and tumors (Laule et al., 2017). The power to resolve and individually characterize the different components can be boosted by combining multiple relaxation- and diffusion-encoding blocks and analyzing the data as joint probability distributions of the relevant observables (English et al., 1991). These ideas follow the principles of multidimensional nuclear magnetic resonance (NMR) spectroscopy and form the basis for multidimensional Laplace NMR

which has become routine in the field of porous media (Galvosas and Callaghan, 2010;Song, 2013) and is now being combined with MRI (Zhang and Blumich, 2014;Benjamini and Basser, 2017). Recently, similar relaxation-diffusion correlation protocols have been translated to *in vivo* studies using model-based rather than nonparametric data inversion (De Santis et al., 2016;Veraart et al., 2017). So far, relaxation-diffusion correlation studies have relied on the Stejskal-Tanner experiment (Stejskal and Tanner, 1965), a pulsed gradient spin-echo (PGSE) sequence that has been in use for more than 50

years and where the signal is encoded for diffusion along a single axis using a pair of collinear gradient pulses. The limitations of the conventional experimental design become apparent when considering a white matter voxel comprising anisotropic domains with multiple orientations. When projected onto the measurement axis defined by the magnetic field gradients, the combination of diffusion anisotropy and orientation dispersion gives rise to a broad distribution of effective diffusivities (Topgaard and Söderman, 2002) that is challenging to retrieve with nonparametric Laplace inversion and, most

importantly, impossible to differentiate from a spread of isotropic diffusivities (Mitra, 1995). Consequently, despite the fact that the relaxation-diffusion correlation yields more detailed information than conventional quantitative MRI, the inherent limitations of the Stejskal-Tanner experiment prevent unambiguous discrimination between isotropic and anisotropic contributions to the diffusivity distributions as well as model-free resolution of tissue microenvironments for heterogeneous anisotropic materials such as brain tissue.

We have recently shown that data acquisition and processing schemes for correlating isotropic and anisotropic nuclear interactions in multidimensional solid-state NMR spectroscopy (Schmidt-Rohr and Spiess, 1994) can be translated to diffusion NMR (de Almeida Martins and Topgaard, 2016), relaxation-diffusion correlation NMR (de Almeida Martins and Topgaard, 2018), and diffusion MRI (Topgaard, 2019), yielding nonparametric diffusion tensor distributions (Jian et al., 2007) with resolution of multiple isotropic and anisotropic diffusion components. These "multidimensional diffusion MRI"

methods (Topgaard, 2017) rely on varying both the amplitude and orientation of the magnetic field gradients within a single encoding block in order to mimic the effects of sample reorientation (Frydman et al., 1992) and rotor-synchronized radio frequency pulse sequences (Gan, 1992) in multidimensional solid-state NMR to target specific aspects of the tensorial property being investigated. Here, we incorporate these ideas into a clinically feasible relaxation-diffusion correlation MRI protocol to quantify the microscopic heterogeneity of the living human brain. The suggested acquisition and analysis

protocols resolve tissue heterogeneity on a five-dimensional space of transverse relaxation rates and axisymmetric diffusion tensors that report on the underlying chemical composition and microscopic geometry. Nonparametric relaxation-diffusion distributions are obtained for each voxel in the three-dimensional image using Monte Carlo data inversion to deal with the non-uniqueness of the Laplace inversion and estimate the uncertainty of quantitative parameters derived from the distributions (Prange and Song, 2009). Sub-voxel tissue environments are resolved without limiting assumptions on the

number or properties of the individual components and characterized with statistical measures that have intuitive relations with the local microstructure.

## 2 Methods

### 2.1 Multidimensional relaxation-diffusion encoding

**Figure 1A** displays a pulse sequence wherein the signal $S(\tau_E, \mathbf{b})$ from a given voxel is encoded for information about the

transverse relaxation rate $R_2$ ($R_2 = 1/T_2$ where $T_2$ is the transverse relaxation time) and diffusion tensor $\mathbf{D}$ by the experimental variables echo time $\tau_E$ and diffusion encoding tensor $\mathbf{b}$ according to (de Almeida Martins and Topgaard, 2018)

$$\frac{S(\tau_E, \mathbf{b})}{S_0} = \int_0^{+\infty} \int_{\mathbf{D} \in \mathrm{Sym}_3^+} P(R_2, \mathbf{D})\, K(\tau_E, \mathbf{b}, R_2, \mathbf{D})\, \mathrm{d}\mathbf{D}\, \mathrm{d}R_2\ , \tag{1}$$

where $P(R_2, \mathbf{D})$ is a joint probability distribution of $R_2$ and $\mathbf{D}$, the kernel $K(\tau_E, \mathbf{b}, R_2, \mathbf{D})$ links the analysis space $(R_2, \mathbf{D})$ to the acquisition space $(\tau_E, \mathbf{b})$, $S_0$ denotes the signal amplitude at $(\tau_E = 0, \mathbf{b} = 0)$, and $\mathrm{Sym}_3^+$ represents the mathematical space containing all 3×3 symmetric positive-definite matrices. The magnetic field gradient waveforms define an axially-symmetric

$b$-tensor that is parameterized by its trace ($b$), orientation ($\Theta, \Phi$), and normalized anisotropy ($b_\Delta$) (Eriksson et al., 2015), the latter controlling the influence of diffusion anisotropy on the detected signal in a manner corresponding to the effect of the angle between the main magnetic field and the rotor spinning axis in solid-state NMR (Frydman et al., 1992). While diffusion encoding performed by a conventional PGSE sequence is limited to a single $b$-tensor "shape" ($b_\Delta = 1$), we have

shown that variation of $b_\Delta$ enables model-free separation and quantification of the isotropic and anisotropic contributions to
the diffusion tensors (de Almeida Martins and Topgaard, 2016). In this work, we used the numerically optimized gradient
waveforms displayed in **Figure 1B** (Sjölund et al., 2015) to generate $b$-tensors at four distinct values of $b_\Delta$. In common with
conventional diffusion MRI, our method requires a minimum echo time of $\sim 50$ ms to accommodate diffusion encoding,
causing the signal contributions from components with $R_2 > 60$ s$^{-1}$ to be reduced to less than 5% of their initial amplitude.
This means that the proposed protocol would require substantial signal averaging in order to quantify the fractions of fast
relaxing components, thus precluding a mapping of myelin water ($R_2 \approx 70$ s$^{-1}$) – one of the primary focuses of early multi-
echo MRI methods (Mackay et al., 1994) – within a time compatible with either clinical practice or research.

Throughout the signal encoding process, the relaxation and diffusion of water are both affected by molecular exchange
between chemically different sites and interactions with cell membranes. Averaging all these complex effects into sets of
effective relaxation rates and apparent diffusion tensors, sub-voxel composition can be reported as a collection of
independent tissue microenvironments, each of which is characterized by a set of ($R_2$,**D**) coordinates (de Almeida Martins
and Topgaard, 2018). Assuming axial symmetry, the various microscopic diffusion tensors are parameterized by four
independent dimensions: two eigenvalues corresponding to the axial and radial diffusivities, $D_\parallel$ and $D_\perp$, and the polar and
azimuthal angles, $\theta$ and $\phi$, describing the orientation of **D** relative to the laboratory frame of reference. The $D_\parallel$ and $D_\perp$
diffusivities can be combined to define measures of isotropic diffusivity, $D_{iso} = (D_\parallel + 2D_\perp)/3$, and normalized diffusion
anisotropy, $D_\Delta = (D_\parallel - D_\perp)/3D_{iso}$ (Eriksson et al., 2015), which report on the "size" and "shape" of the corresponding
microscopic diffusion patterns (Topgaard, 2017). Tissue microscopic heterogeneity is therefore characterized with
$P(R_2,D_{iso},D_\Delta,\theta,\phi)$ distributions, whose dimensions directly correspond to those of the 5D acquisition space ($\tau_E,b,b_\Delta,\Theta,\Phi$):

$$\frac{S\left(\tau_E,b,b_\Delta,\Theta,\Phi\right)}{S_0} = \int_0^\infty \int_0^\infty \int_{-1/2}^1 \int_0^\pi \int_0^{2\pi} K\left(\tau_E,b,b_\Delta,\Theta,\Phi,R_2,D_{iso},D_\Delta,\theta,\phi\right)$$
$$\times P\left(R_2,D_{iso},D_\Delta,\theta,\phi\right)\mathrm{d}\phi\sin\theta\ \mathrm{d}\theta\ \mathrm{d}D_\Delta \mathrm{d}D_{iso}\ \mathrm{d}R_2. \tag{2}$$

The relaxation-diffusion encoding kernel is defined as

$$K(...) = \exp\left(-\tau_E R_2\right)\exp\left(-bD_{iso}\left[1+2b_\Delta D_\Delta P_2\left(\cos\beta\right)\right]\right), \tag{3}$$

where $P_2(x) = (3x^2-1)/2$ denotes the 2nd Legendre polynomial, and $\beta$ is the arc-angle between the major symmetry axes of **b**
and **D**, given by $\cos\beta = \cos\Theta\cos\theta + \cos(\Phi-\phi)\sin\Theta\sin\theta$. According to Eq. (3), each ($\tau_E,b,b_\Delta,\Theta,\Phi$) coordinate establishes
correlations across the separate dimensions of the $R_2$-**D** space. Consequently, sampling various combinations of echo times
and $b$-tensor parameters facilitates a comprehensive mapping of tissue-specific relaxation and diffusion properties.

## 2.2 MRI measurements

A healthy volunteer (female, 31 years) was scanned on a Siemens Magnetom Prisma 3T system equipped with a 20-channel receiver head-coil, and capable of delivering gradients of 80 mT/m at the maximum slew rate of 200 T/(m·s). The measurements were approved by a local Institutional Review Board (Partners Healthcare System), and the research subject provided written informed consent prior to participation.

Experimental data were acquired using the prototype spin-echo sequence (Lasič et al., 2014) and gradient waveforms shown in **Figure 1**. The depicted waveforms give four distinct $b$-tensor anisotropies ($b_\Delta = \{-0.5, 0.0, 0.5, 1.0\}$), which were probed at varying combinations of echo-times, $b$-values, and $b$-tensor orientations. The waveforms giving $b_\Delta = -0.5$, 0.0, and 0.5 (see **Figure 1B**) were calculated with a numerical optimization package (Sjölund et al., 2015) (https://github.com/jsjol/NOW), including compensation for the effects of concomitant gradients (Szczepankiewicz et al.). This procedure yielded a pair of asymmetric gradient waveforms lasting 30.8 ms and 25.0 ms, separated by approximately 8.0 ms. Linear encoding ($b_\Delta = 1$) was implemented with two separate gradient waveforms; a symmetric bipolar gradient waveform whose encoding blocks lasted $\tau = 25.1$ ms and were separated by 8.0 ms (see **Figure 1B**), and a pair of $\tau = 15.1$ ms single-pulsed gradients bracketing a time-period of 13.7 ms. The spectral profile of the bipolar gradient waveform was tuned to that of the asymmetric gradient waveforms in order to reduce the influence of time-dependent diffusion (Woessner, 1963;Callaghan and Stepišnik, 1996).

A total of 852 individual images were recorded at different combinations of ($\tau_E,b,b_\Delta,\Theta,\Phi$) throughout the entire scan time of 45 minutes. The acquisition protocol is summarized in **Figure 2A**. Briefly, $b_\Delta = 1$ was acquired over 72 directions distributed over four $b$-values (6, 10, 16, and 40 directions at $b = 0.1$, 0.7, 1.4, and $2\cdot10^9$ sm$^{-2}$, respectively), both $b_\Delta = -0.5$, and 0.5 were collected across 64 directions spread out over four $b$-values (6, 10, 16, and 32 directions at, respectively, $b = 0.1$, 0.7, 1.4, and $2\cdot10^9$ sm$^{-2}$), and $b_\Delta = 0$ was acquired for a single gradient waveform orientation, repeated 6 times over six $b$-values ($b = 0.1$, 0.3, 0.7, 1, 1.4, and $2\cdot10^9$ sm$^{-2}$). For each ($b,b_\Delta$) coordinate, the set of directions was optimized using an electrostatic repulsion scheme (Bak and Nielsen, 1997;Jones et al., 1999). The various ($b,b_\Delta,\Theta,\Phi$) sets were then repeatedly acquired at three different echo-times ($\tau_E = 80$, 110, and 150 ms) using the spectrally-tuned waveforms. The non-tuned Stejskal-Tanner waveform was used to acquire $b_\Delta = 1$ data at $\tau_E = 60$ and 80 ms. Comparison between data acquired with the bipolar and the Stejskal-Tanner gradient waveforms at $\tau_E = 80$ ms allowed us to assess the validity of the Gaussian diffusion approximation (Callaghan and Stepišnik, 1996).

All images were recorded using a repetition time of 3 s, and an echo-planar readout with a 220×220×66 mm$^3$ field of view, spatial resolution of 2×2×6 mm$^3$, and a partial Fourier factor of 6/8. Spatial resolution was sacrificed in favor of high signal-to-noise ratios (SNR). The 2×2×6 mm$^3$ anisotropic voxel configuration enables a large coverage with a minimal number of slices and yields axial maps with a high spatial resolution wherein anatomical features of interest can be easily identified. The acquired images were corrected for subject motion in *ElastiX* (Klein et al., 2009), using the extrapolated reference

method detailed in (Nilsson et al., 2015). Motion-corrected and non-motion-corrected data were then inverted using a quick

12-bootstrap procedure (see the following sub-section for more details on the inversion), and the resulting parameter maps were subsequently compared. As no substantial differences were found between the results from the corrected and non-corrected datasets, we opted to not use motion correction in our final analysis. No denoising approaches were used prior to data inversion.

## 2.3 Nonparametric Monte Carlo inversion


Algorithms designed to solve Eq. (2) have been reviewed in both general (Istratov and Vyvenko, 1999) and magnetic resonance (Mitchell et al., 2012) literature. While classical inversion methods can be successfully used to estimate the 5D $P(R_2,D_{iso},D_{\Delta},\theta,\phi)$ distribution, they become memory costly at the high dimensionality of our protocol. To circumvent this difficulty, we introduced an inversion approach wherein our correlation space is explored through a directed iterative

algorithm, as explained in (de Almeida Martins and Topgaard, 2018). The algorithm starts by randomly selecting 200 points from the $(0 < \log(R_2/\text{s}^{-1}) < 1.5, -10 < \log(D_{\parallel}/\text{m}^2\text{s}^{-1}) < -8.5, -10 < \log(D_{\perp}/\text{m}^2\text{s}^{-1}) < -8.5, 0 < \cos\theta < 1, 0 < \phi < 2\pi)$ space. A discrete $P(R_2,\mathbf{D})$ distribution is then estimated by solving a discretized version of Eq. (2) via a standard non-negative least squares (NNLS) algorithm (Lawson and Hanson, 1974). Points with non-zero weights are stored, merged with a new randomly-generated set of 200 $(R_2,D_{\parallel},D_{\perp},\theta,\phi)$ points, and the weights of the merged set of points are found through a NNLS

fit (Lawson and Hanson, 1974). The process of selecting points with non-zero weights, subsequently merging them with a random $(R_2,D_{\parallel},D_{\perp},\theta,\phi)$ configuration, and finally fitting the merged set is repeated for a total of 20 times in order to find a $P(R_2,D_{\parallel},D_{\perp},\theta,\phi)$ distribution yielding a low residual sum of squares. Following 20 rounds, the resulting $(R_2,D_{\parallel},D_{\perp},\theta,\phi)$ configuration is selected, split, and subjected to a small random mutation. The original and mutated configurations are merged and a new $P(R_2,D_{\parallel},D_{\perp},\theta,\phi)$ distribution is determined by fitting the merged set to the data using the NNLS algorithm

(Lawson and Hanson, 1974). The mutation and fitting procedure is repeated 20 times to find the local $(R_2,D_{\parallel},D_{\perp},\theta,\phi)$ configuration corresponding to the lowest sum of squared residuals. A final plausible $P(R_2,D_{\parallel},D_{\perp},\theta,\phi)$ solution is subsequently estimated at the end of the mutation cycle by selecting the 10 $(R_2,D_{\parallel},D_{\perp},\theta,\phi)$ points with the highest weights and performing a final NNLS fit.

The procedure described above is performed voxel-wise, resulting in an array of spatially resolved $P(R_2,D_{\parallel},D_{\perp},\theta,\phi)$ discrete

distributions. Owing to the stochastic nature of the inversion protocol, we may fail at retrieving a non-trivial solution, which produces a small number of randomly located black voxels in the parameter maps. To correct for this, we combine the points from each voxel with the ones from its six nearest-neighbors, subsequently fitting the set of 7×10 points to the underlying signal in order to find the 10 most likely points. The new $(R_2,D_{\parallel},D_{\perp},\theta,\phi)$ set is fitted to the signal, and the resulting $P(R_2,\mathbf{D})$ is taken as the solution of the analyzed voxel. Finally, the $P(R_2,D_{\parallel},D_{\perp},\theta,\phi)$ distribution is mapped onto the $(R_2,D_{iso},D_{\Delta},\theta,\phi)$

space.

Following the works of Prange and Song (Prange and Song, 2009), we replace traditional regularization constraints (Whittall and MacKay, 1989) with an unconstrained Monte Carlo approach that estimates voxel-wise ensembles of $N$ distinct $P(R_2,\mathbf{D})$ solutions consistent with the primary data (de Almeida Martins and Topgaard, 2018). In this study, we estimated ensembles of $N = 96$ solutions per voxel. The level of dispersion within a given solution set characterizes the uncertainty of the inversion procedure, and can thus be used to estimate the uncertainty of any quantities derived from $P(R_2,\mathbf{D})$ (Prange and Song, 2009;de Almeida Martins and Topgaard, 2018).

The nonparametric Monte Carlo inversion procedure was implemented in MATLAB and is publicly available in our GitHub repository https://github.com/JoaoPdAMartins/md-dmri (Nilsson et al., 2018b). Inversion of the 45 minutes dataset took ~72 hours on a 12-Core Intel Xeon E5 2.7-GHz CPU, with a 64-GB DDR3 memory.

## 3 Results

### 3.1 Spatially-resolved 5D relaxation-diffusion distributions

The proposed acquisition protocol translates into distinctive signal decay curves for each of the main components of the human brain. Indeed, voxels encompassing either white matter WM, gray matter GM, or cerebrospinal fluid CSF, are all characterized by clearly distinct signal patterns (see **Figure 2B**). The observed differences can be used to infer the gross $R_2$-$\mathbf{D}$ properties of the various cerebral constituents: WM signals are highly sensitive to both $b_\Delta$ and $(\Theta,\Phi)$, indicative of anisotropic diffusion along coherently aligned microscopic domains; GM signal patterns are rather insensitive to $b_\Delta$ and $(\Theta,\Phi)$, consistent with isotropic diffusion; and CSF data decays quickly with increasing $b$ while remaining mostly unaffected by the other acquisition variables, features that suggest an isotropic medium characterized by relatively low $R_2$-values. Voxels comprising mixtures of WM, GM, and/or CSF generate patterns that can be interpreted as a superposition of the signal data from the pure components.

Spatially resolved 5D $R_2$-$\mathbf{D}$ nonparametric distributions are retrieved from the experimental data using the model-free inversion approach described in the Methods section. **Figure 2C** displays the solution ensembles for voxels containing WM, GM, and CSF, as well as combinations of those components: WM+GM, WM+CSF, and GM+CSF. Brain tissue possesses various microscopic components, whose relaxation and diffusion properties differ over various orders of magnitude. Therefore, tissue heterogeneity is more suitably described with logarithmic distributions, where pore anisotropy is parameterized with $\log(D_\parallel/D_\perp)$ instead of $D_\Delta$. The distinctive characters of the raw signal patterns in **Figure 2B** result in unique voxel-wise distributions that capture the gross microscopic features of the main cerebral components. Namely, CSF is characterized by high $D_{iso}$, low $R_2$, and $D_\parallel \sim D_\perp$; in contrast, GM and WM both exhibit lower $D_{iso}$ and higher $R_2$, with WM being differentiated by its high $D_\parallel/D_\perp$. As expected, voxels comprising mixtures of WM, GM, and CSF yield a linear combination of the distributions from the individual components.

Voxels containing pure GM or WM are characterized by clusters of $P(R_2,\mathbf{D})$ components covering a significant range of the $R_2$-$\mathbf{D}$ space. Because both tissue types comprise a plethora of cells with varying geometries or chemical compositions (*e.g.* axons with various amounts of myelin, dendrites, or glial cells), the observed spread may be interpreted as a direct consequence of the underlying cellular heterogeneity. However, similar broad distributions were also observed in spectroscopic multidimensional diffusion correlation measurements of discrete-component phantoms (de Almeida Martins and Topgaard, 2016, 2018), hinting that the solution spread additionally reflects the measurement and inversion uncertainty. This intrinsic uncertainty masks the effects of finer cellular details like the intra- and extra-axonal components modeled in previous diffusion-relaxation correlation MRI methods (Veraart et al., 2017).

As evidenced by **Figure 2C**, pure GM voxels yield bimodal distributions that feature a nearly symmetric spread of components around the $\log(D_\parallel/D_\perp) = 0$ plane. The bimodality of the GM distributions is an artefact attributed to the fact that prolate ($D_\Delta > 0$, $D_\parallel/D_\perp > 1$) and oblate ($D_\Delta < 0$, $D_\parallel/D_\perp < 1$) diffusion tensors with similar $D_\mathrm{iso}$ yield signal patterns that are only clearly discerned when $D_\Delta > 0.5$ or, equivalently, $D_\parallel/D_\perp > 4$ (Eriksson et al., 2015). Diffusion tensor imaging (DTI) studies of the human cortex have revealed a low, yet non-negligible, diffusion anisotropy in cortical GM tissue (Assaf, 2018). The observation of both oblate and prolate components in the pure GM voxel is consistent with those findings, with the intrinsically low anisotropy preventing an unambiguous distinction between $D_\Delta > 0$ or $D_\Delta < 0$ solutions. The artefactual spread of anisotropic components is expected to worsen with the increase of experimental noise. Random signal fluctuations create small differences between data acquired at different $b_\Delta$-values, and consequently introduce a preference for anisotropic components with arbitrary $D_\Delta$ sign. This effect is similar to the "eigenvalue repulsion" artefact in conventional DTI, where noise introduces a discrepancy in the eigenvalues of the voxel-averaged diffusion tensor that in turn gives rise to a positive bias in anisotropy (Pierpaoli and Basser, 1996;Jones and Cercignani, 2010).

### 3.2 Statistical measures of tissue heterogeneity

The $R_2$-$\mathbf{D}$ distribution ensembles provide a wealth of information that is challenging to visualize in spatially resolved datasets with large image matrices. Drawing inspiration from the field of porous media, where ensembles of distributions have been converted into ensembles of scalar parameters such as total porosity or fraction of bound fluid (Prange and Song, 2009), we extract statistical measures from the $R_2$-$\mathbf{D}$ distributions. A multitude of statistical functionals can be computed from the same distribution, meaning that the per-voxel $P(R_2,\mathbf{D})$ ensembles generate a comprehensive set of distinct voxel-wise parameters. As shown in **Figure 3**, the Monte Carlo realizations of $P(R_2,\mathbf{D})$ are translated into ensembles of statistical measures, with 96 individual estimates being extracted for each measure. For compactness, the ensembles of statistical parameters are reduced to an average $\langle \cdot \rangle$ and a dispersion measure $\sigma[\cdot]$ that is interpreted as the uncertainty of the estimated functional (Prange and Song, 2009). To render the results more robust to outliers, we report $\langle \cdot \rangle$ as the ensemble median and estimate $\sigma[\cdot]$ as a median absolute deviation. The calculation of averages (as measured by the median) reduces the underlying ensemble of solutions into a single scalar, and allows us to convey intra-voxel composition with parameter maps

of average mean values $\langle E[x] \rangle$, average variances $\langle Var[x] \rangle$ and average covariances $\langle Cov[x,y] \rangle$ of all the relevant dimensions of the 5D $R_2$-**D** space (see **Figure 3**). All of the statistical measures derived in this work parameterize diffusion tensor anisotropy with $D_\Delta^2$ rather than $D_\Delta$; this is motivated by the intrinsic difficulty of distinguishing between prolate and oblate tensors (Eriksson et al., 2015).

The three maps in the first column of **Figure 3** provide a rough spatial overview of the principal tissue types: $\langle E[R_2] \rangle$ and $\langle E[D_\text{iso}] \rangle$ clearly identify CSF-rich areas (low $\langle E[R_2] \rangle$ and high $\langle E[D_\text{iso}] \rangle$), while high $\langle E[D_\Delta^2] \rangle$ values separate WM from the two other main cerebral tissues. However, mean parameter maps alone cannot identify or characterize intra-voxel heterogeneity, and their use should be complemented with dispersion measures including, but not limited to, the (co)variance elements displayed in columns 2 and 3 of **Figure 3**. For example, voxels surrounding the ventricles do not show a truly distinctive feature in maps of mean values but are characterized by non-zero covariance matrix elements. To understand the origin of the non-zero values, let us focus on the WM+CSF and GM+CSF voxels indicated in **Figure 3**. The corresponding $P(R_2,\mathbf{D})$ distributions (displayed in **Figure 2C**) comprise two populations at distant $(R_2,D_\text{iso})$ coordinates, and both voxels are thus characterized by high values of $Var[R_2]$ and $Var[D_\text{iso}]$ (see histograms of **Figure 3**). As CSF and GM are both characterized by a low anisotropy, GM+CSF exhibits low values of $Var[D_\Delta^2]$; in contrast, WM+CSF displays a significant dispersion along $D_\Delta^2$, which results in high $Var[D_\Delta^2]$ values. Covariance measures inform about the correlations across the various dimensions of the $R_2$-**D** space. In WM+CSF distributions, for instance, higher values of diffusion anisotropy are correlated with higher $R_2$ and lower $D_\text{iso}$, which results in positive $Cov[R_2,D_\Delta^2]$ and negative $Cov[D_\text{iso},D_\Delta^2]$. The elevated $\langle Var[R_2] \rangle$ and $\langle Var[D_\text{iso}] \rangle$, and negative $\langle Cov[R_2,D_\text{iso}] \rangle$ values found in the ventricular regions are thus interpreted as a product of sub-voxel combinations of CSF with other components. A combination of high $\langle Var[D_\Delta^2] \rangle$, positive $\langle Cov[R_2,D_\Delta^2] \rangle$, and negative $\langle Cov[D_\text{iso},D_\Delta^2] \rangle$ locate WM+CSF voxels in those same regions, while low values of $\langle Var[D_\Delta^2] \rangle$ indicate the existence of deep gray matter in the vicinity of the ventricles.

The maps displayed in **Figure 3** can also be used to identify voxels containing WM+GM mixtures. Because WM and GM distributions are characterized by similar values of $R_2$ and $D_\text{iso}$, WM+GM voxels result in nearly zero values of $Var[R_2]$, $Var[D_\text{iso}]$, $Cov[D_\text{iso},y]$ and $Cov[R_2,y]$. Instead, WM+GM voxels are signaled by finite values of $\langle Var[D_\Delta^2] \rangle$, originated by the $\log(D_\parallel/D_\perp)$ spread observed in the underlying $R_2$-**D** distribution (see the WM+GM distribution in **Figure 3C**).

### 3.3 Bin-resolved metrics of tissue heterogeneity

A more detailed picture of intra-voxel heterogeneity is obtained by dividing the distribution space into smaller subspaces ('bins'). In line with early diffusion MRI works (Pierpaoli et al., 1996), we define three bins that loosely capture the diffusion properties of the $P(R_2,\mathbf{D})$ distributions from the main brain components (see Table 1 and **Figure 4A**). The 'Big' bin contains CSF contributions, whereas the 'Thin' and 'Thick' bins capture the signal fractions from WM and GM, respectively. The names 'Big', 'Thin', and 'Thick' are inspired by the geometric properties of the microscopic diffusion

tensors that are captured by each individual bin. Visual inspection of **Figure 4B** reveals that the spatial distributions of the three bins are consistent with the expected distributions of the corresponding tissues, providing more evidence that the coarsely defined bins allow a separation of the main cerebral constituents. Parameter maps of the per-bin means of the relaxation and diffusion properties are more straightforwardly interpreted than the heterogeneity measures derived from the entire distribution space: for example, the deep gray matter inferred in the previous paragraph is easily identifiable at the center (white arrows) of the 'Thick' maps of **Figure 4B**. Further, the correlations across the various dimensions of the diffusion space allow the resolution of subtle differences in relaxation rates. Focusing on the first column of **Figure 4B**, we notice that the 'Thick' fraction exhibits a slightly lower $R_2$ rate than that of the 'Thin' fraction. This behavior is in accordance with previous literature (Tofts, 2003) and is consistently observed across the entire slice.

Global and bin-resolved averages for all the analyzed voxels of the entire 3D image matrix are compiled in **Figure 5**, where per-voxel average means of $R_2$, $D_{iso}$, and $D_\Delta^2$ are plotted against their respective uncertainties, $\sigma[E[R_2]]$, $\sigma[E[D_{iso}]]$, and $\sigma[E[D_\Delta^2]]$, and average signal amplitudes $\langle S_0 \rangle$. Although the displayed statistical analysis is restricted to mean values, similar calculations can be done using any other scalar measure derived from the 5D $R_2$-**D** distributions. Examination of the scatter plots in **Figure 5** shows that microscopic populations with low signal fractions generate statistical measures with significantly higher uncertainties. While no immediate correlation is discerned between the estimated mean values and their corresponding uncertainty, the negative correlation between uncertainty and signal fractions introduces a significant dispersion of $\langle E[x] \rangle$ at $\langle S_0 \rangle / \max(\langle S_0 \rangle) < 0.1$ (see, for example, the $D_{iso}$ scatterplots for the 'Thin' and 'Thick' populations). Despite the lower precision at low $\langle S_0 \rangle$, the various average mean values are observed to be nearly constant throughout the $\langle S_0 \rangle / \max(\langle S_0 \rangle) > 0.1$ region; the only exception is $\langle E[D_\Delta^2] \rangle$ for the 'Thin' fraction, which shows a higher susceptibility to noise as evidenced by its positive correlation with $\langle S_0 \rangle$.

The minor differences between the relaxation rates of the 'Thin' and 'Thick' components are also observed in the scatter plots of **Figure 5**. A more detailed analysis shows that distinct $R_2$-rates can be consistently detected in voxels containing GM+WM mixtures (see **Figure 6A**), where conventional 1D $R_2$ distributions fail to resolve the subtle differences between components (Whittall et al., 1997). The second and third columns of **Figure 6A** display mixed voxels, where the 'Thin' and 'Thick' populations each account for at least 30% of the total measured signal. Approximately 75% of the mixed voxels exhibit $R_2$ differences greater than the estimated uncertainties, thus providing evidence that the differentiation between the $R_2$-rates of the two bins is indeed a meaningful result.

All bin-resolved $\langle E[R_2] \rangle$ plots in **Figure 5** display a secondary cluster at high $R_2$-values. Inspection of **Figure 6B** reveals that the fast relaxing cluster corresponds to the non-masked extra-meningeal tissues and, for the 'Thin' fraction, to the pallidum (region 1 in **Figure 6B**), a major component of the basal ganglia structures located deep in the brain. The contributions from the high-$R_2$ components are observed to concentrate around $R_2 = 30$ s$^{-1}$ (*cf*. **Figure 6A**), the upper $R_2$-limit of the Monte Carlo inversion procedure. The 'pile-up' of fast-relaxing contributions around the maximum allowed $R_2$ value is a well-known artefact of Laplace inversions (Saab et al., 1999).

The $\langle E[R_2] \rangle$ map of the 'Thick' bin features three main $R_2$ populations: high $R_2$ in the skull region (red voxels), low $R_2$ in peripheral brain regions (green voxels), and intermediate $R_2$ values in the inner brain regions (yellow voxels). To more easily inspect the spatial distribution of the various populations within the 'Thick' bin we delimited the ($-3.5 < \log(D_\parallel/D_\perp) < 0.6$, $-10 < \log(D_{iso}/\text{m}^2\text{s}^{-1}) < -8.7$) sub-space in three separate $R_2$ regions, and defined the 'Low' ($-0.5 < \log(R_2/\text{s}^{-1}) < 1.2$), 'Medium' ($1.2 < \log(R_2/\text{s}^{-1}) < 1.4$), and 'High' ($1.4 < \log(R_2/\text{s}^{-1}) < 2$) sub-bins of **Figure 6C**. In $T_2$ units, the 'Low', 'Medium', and 'High' bins correspond to 63 ms to 3.16 s, 40 ms to 63 ms, and 10 ms to 40 ms. Note that the true upper boundary of the 'High' bin is set by the limits of the Monte Carlo inversion and is equal to $R_2 = 30$ s$^{-1}$; the $R_2 = 100$ s$^{-1}$ boundary is defined simply to render a more aesthetically pleasing plot (see **Figure 6C**). The bin-resolved signal fraction maps were then compared with a high-resolution longitudinal relaxation ($R_1$) weighted image segmented in four tissue classes: WM, cortical GM, deep GM, and CSF. **Figure 6C** shows that the spatial distributions of the 'Low', and 'Medium' sub-fractions roughly correspond to the expected distributions of cortical GM, and deep GM structures, respectively. Despite the similarities between bin-resolved and segmentation maps, the former possesses a grainier appearance and seem to miss a significant portion of deep GM tissue at the center of the slice. While the grainier aspect is caused by the higher noise of the $R_2$-**D** correlation dataset, the absence of central GM is explained by the presence of anisotropic tissues in structures such as the pallidum (region 1 in **Figure 6B**) and the thalamus (region 2 in **Figure 6B**). Those two deep GM structures are then contained within the 'Thin' bin, and not within the 'Thick' bin from which we defined the $R_2$ sub-spaces. Joining the contributions of cortical and deep GM within a single tissue class offers further insight on the link between microscopic tissue composition and binning (see **Figure 6D**). Comparing the 3-tissue segmentation with maps of the 'Big', 'Thin', and 'Thick' fractions confirms that the pallidum and part of the thalamus are captured by the 'Thin' bin.

### 3.4 Clinical feasibility of the $R_2$-D correlation approach

The acquisition protocol discussed thus far can be inserted without further alterations in research studies of brain disease, where subjects are recruited for long scan sessions. However, the associated 45 min scan time impedes its use outside of a clinic-research setting. To assess the potential for clinical translation of the proposed framework, we compare the performance of the exhaustive 45 min protocol with that of an abbreviated protocol, compatible with the time frame of most clinical applications. To this end, we included two different 5D relaxation-diffusion MRI protocols in a single imaging session: the 45 min protocol described in the *Methods* section, and an abbreviated 15 min protocol whose details are contained in the *Supplementary Material*. The two acquisition protocols were consecutively used without repositioning the volunteer.

The abbreviated dataset was inverted with the Monte Carlo algorithm described above. The resulting 5D $R_2$-**D** distributions and parameter maps are compiled in the *Supplementary Material*. **Figure 7** shows the bin-resolved parameter maps obtained with the 15 min acquisition protocol. Overall, the parameter maps derived from the abbreviated data resemble slightly noisier reproductions of the maps computed from the exhaustive protocol and provide the same conclusions. Namely, the

'Big', 'Thin', and 'Thick' bins demarcate the signal contributions from CSF, WM, and GM, respectively, and the main $R_2$-**D**

properties of those same tissue types are accurately captured by the per-bin mean parameter maps. The most obvious difference between the two datasets is the lower quality of the $R_2$ metrics derived from the abbreviated data. This is evidenced by unreasonably high $R_2$-rates in the ventricles (see the $\langle E[R_2] \rangle$ maps in **Figure 7B**), and a higher difficulty in separating between the mean $R_2$-rates of the 'Thin' and 'Thick' bins. Only 65% of mixed voxels from the abbreviated dataset show a meaningful $R_2$ separation, as opposed to the 75% determined in the previous sub-section. The lower resolution along

the $R_2$-dimension is most likely explained by the fact that the abbreviated protocol concentrates 85% of its measurements within two unique values of $\tau_E$, an acquisition scheme that is quite unspecific to dispersion along $R_2$. In future experiments, we plan to address this issue by enforcing a more uniform distribution of data points along the various echo-times.

## 4 Discussion and Conclusions

The proposed framework resolves intra-voxel heterogeneity on a 5D space of transverse relaxation rates $R_2$ and diffusion

tensor parameters $(D_{\text{iso}}, D_\Delta, \theta, \phi)$. Per-voxel brain composition is broken down into a non-predefined number of microscopic environments with clearly distinct relaxation and diffusion properties. The heterogeneity within a voxel is thus resolved as linear combinations of independent microscopic components that can be assigned to local tissue environments; on a global scale, the sub-voxel environments can be grouped into more general tissue classes. For healthy brain tissue, the detected microenvironments were classified into three broad bins whose diffusion properties respectively match those of the main

constituents of the brain: WM, GM, and CSF. The separation between contributions from the three bins was observed to provide a clean 3D mapping of WM, GM, and CSF that agrees well with a conventional $R_1$-based tissue segmentation. This demonstrates that the proposed protocol can indeed separate between sub-voxel tissue environments with different relaxation and diffusion properties; in the healthy human brain, the resolved environments can be coarsely assigned to contributions from CSF, WM, and GM (see **Figure 6D**). The distinction between microscopic tissue environments with different $R_2$-**D**

properties provides complementary information to $R_1$-weighted segmentation and enables the resolution of tissue heterogeneity within a single anatomical structure, *e.g.*, resolving anisotropic and isotropic regions within the thalamus.

The protocol presented in this work shows promise for neuroanatomy studies dealing with the resolution of specific microscopic features such as nerve fiber-tracking through heterogeneous voxels (Jeurissen et al., 2014) or free water mapping (Pasternak et al., 2009). Within a clinical setting, disentangling different tissue signals is expected to be useful for

pathological conditions associated with intra-voxel tissue heterogeneity, *e.g.* tumor infiltration in surrounding brain tissue, inflammation of cerebral tissue, or replacement of myelin with free water. In the latter example, the proposed echo-times lead to an almost complete decay of the signal contributions from myelin domains, meaning that the effects of axonal demyelination would have to be probed indirectly by tracking a reduction of the signal fraction from anisotropic sub-voxel components.

Several approaches have been introduced in the diffusion MRI literature where sub-voxel composition is investigated by devising signal models with increasingly complex priors and constraints (Wang et al., 2011;Zhang et al., 2012;Scherrer et al., 2016). While such models can be used to investigate the conditions mentioned in the above paragraph, the attained conclusions will be heavily dependent on the assumptions used to construct the model (Novikov et al., 2018). Hence, erroneous conclusions may be derived whenever the presupposed MR properties differ from the underlying microstructure

(Lampinen et al., 2019). This limitation is alleviated in the present framework where sub-voxel heterogeneity is quantified with nonparametric distributions that are retrieved from the data with minimal assumptions on the underlying tissue properties. Moreover, the vast majority of diffusion MRI models has been so far implemented with conventional Stejskal-Tanner sequences, which are known to convolve the signal contributions from $D_\Delta$ and $\mathbf{D}$ orientation. Acquiring data at various $b_\Delta$ has been shown to disentangle between the effects of anisotropy and dispersion in $\mathbf{D}$ orientations (Eriksson et al.,

2013;Eriksson et al., 2015), meaning that our 5D ($\tau_E$,$\mathbf{b}$) acquisition space is expected to provide a more clear component resolution whenever orientation dispersion is present.

Besides resolving the various microscopic domains within a voxel, we were also capable of observing subtle differences in component-specific relaxation rates. As mentioned before, this information is unattainable with classical multi-echo $R_2$ distribution protocols (Whittall et al., 1997), and its extraction leverages on the vast correlations across the full ($D_{iso}$,$D_\Delta$,$\theta$,$\phi$)

space (de Almeida Martins and Topgaard, 2018). We would like to reinforce that small $R_2$ differences can be observed despite the limited number and range of echo-times sampled in this work; here, the separation between $R_2$-components is mostly driven by the excellent resolution in the diffusion dimensions. The measurement of $\mathbf{D}$-resolved transverse relaxation rates may complement previous work on tract-specific $R_1$ rates (De Santis et al., 2016).

At the cellular level, the translational motion of water inside the human brain is influenced by interactions with

macromolecules and partially permeable membranes forming compartments with barrier spacings ranging from nanometers for synaptic vesicles and myelin sheaths to micrometers for the plasma membranes of the axons. The diffusion of water during the 0.1 s time-scale of MRI signal encoding is thus affected by a myriad of complex phenomena that are not explicitly accounted for in Eq. (2). Instead, we use the well-established approach of approximating the micrometer-scale water displacements as a distribution of anisotropic Gaussian contributions (Jian et al., 2007). The measured diffusivities may

depend on the exact choice of experimental variables if the timing parameters of the gradient waveforms match the characteristic time-scales of displacements between cellular barriers (Woessner, 1963) or molecular exchange between tissue environments with distinctly different diffusion properties (Kärger, 1969). By augmenting our acquisition protocol with an experimental dimension in which the spectral profiles of the gradient waveforms are comprehensively varied (Callaghan and Stepišnik, 1996;Lundell et al., 2019), microscopic barrier spacings could in principle be estimated by explicitly including the

effects of restricted diffusion in the kernel of Eq. (2). Here we chose to minimize the influence of time-dependence by designing waveforms with similar gradient-modulation spectra.

In the previous section, we mentioned that prolate ($D_\Delta > 0$) and oblate ($D_\Delta < 0$) diffusion tensors with $|D_\Delta| < 0.5$ result in similar signal decays (Eriksson et al., 2015). In the absence of orientational order, diffusion tensor anisotropy is detected as a deviation from a mono-exponential signal-decay, which, to first order, is proportional to $D_\Delta^2$ (Eriksson et al., 2015).

Consequently, the magnitude of $D_\Delta$ can be easily determined at moderate $b$-values while the sign may require data acquired with $b$-values up to $4 \cdot 10^9$ sm$^{-2}$ (Eriksson et al., 2015) and echo-times comparable to the ones registered in this work; currently, such acquisition parameters can only be achieved with a specialized scanner (Setsompop et al., 2013;Jones et al., 2018).

Resolving and separately characterizing intra- and extra-axonal compartments in brain tissue has been of long-standing

interest in the MRI field (Does, 2018). Recently, Veraart et al. (Veraart et al., 2017) estimated subtle differences in $R_2$ and diffusivity parameters for the intra- and extra-axonal components of human brain white matter by applying a constrained two-component model to data acquired with a conventional relaxation-diffusion correlation protocol relying on the Stejskal-Tanner experiment. The obtained $R_2$-values differ with less than a factor of two while the $D_{iso}$-values are nearly identical and the $D_\Delta$-values are 1 (by constraint) and approximately 0.5 for the intra- and extra-cellular compartments, respectively.

Comparing with the non-parametric distributions in **Figure 2**, we note that components with such similar properties would be virtually impossible to resolve in our minimally constrained approach despite the additional information added by the $b$-tensor shape dimension. The limited resolution is consistent with the fact that Eq. (2) states an ill-posed inverse problem accommodating multiple non-unique solutions – probably also including the one with two 'Thin' components as assumed by Veraart et al. We suggest that the unconstrained inversion could be used as a first analysis tool to define the boundaries of a

more ambitious model incorporating additional information, e.g. from microanatomy studies that is not directly observable in the MRI data.

This work introduces and demonstrates a novel MRI framework, in which the microscopic heterogeneity of the living human brain is characterized via 5D correlations between the transverse relaxation rate $R_2$, isotropic diffusivities $D_{iso}$, normalized diffusion anisotropy $D_\Delta$, and diffusion tensor orientation ($\theta,\phi$). The correlations allow model-free estimation of per-voxel

relaxation-diffusion distributions $P(R_2,\mathbf{D})$ that combine the chemical sensitivity of $R_2$ with the link between microstructure and the diffusion metrics. The rich information content of $P(R_2,\mathbf{D})$ is reported through a set of 21 unique maps obtained by binning and parameter calculation in the 5D distribution space. Being specific to different tissue types while relying on few assumptions, the presented protocol shows promise for explorative neuroscience and clinical studies in which microscopic tissue composition cannot be presumed *a priori*. While the spatial resolution of the data acquired in this work was relatively

limited, sacrificing resolution for SNR, there are several avenues to explore in the future, in hardware, acquisition and analysis that will boost the SNR per unit time, thereby increasing the potential for improved resolution. From the hardware perspective, the use of ultra-high field (7T and above), and ultra-strong field gradients (Setsompop et al., 2013;Jones et al., 2018), can boost SNR and reduce echo-time-per-unit-$b$-value, respectively. For example, as noted in ref. (Jones et al., 2018), for $b_\Delta = 0$ encoding, the shorter $\tau_E$ afforded by stronger gradients such as those available on a Connectom scanner (300

mT/m) results in an improvement in SNR of approximately 50% compared to that achievable on the system used in this study (80 mT/m gradients). From the acquisition perspective, multi-band acquisition schemes (Barth et al., 2016) can speed up overall acquisition times and facilitate a wide brain coverage with smaller voxel-sizes. Moreover, replacing the rectilinear echo-planar readout (Turner et al., 1991) with a spiral read-out (Wilm et al., 2017) can help to further reduce the echo time, boosting SNR which could be traded for higher spatial resolution. From the analysis side, as noted in the *Methods* section, no denoising approaches were applied here. Recent advances in denoising and/or joint reconstruction (Veraart et al., 2016;Bazin et al., 2019;Wang et al., 2019;Haldar et al., 2020) could further enhance the SNR, allowing resolution to be pushed higher. Finally, the presented framework can be merged with MRI fingerprinting methodology (Ma et al., 2013), whose pattern matching algorithms may considerably boost the data inversion speed.

## Data and Code availability

The software analysis tools discussed in this paper are available for downloading from a public GitHub repository: https://github.com/JoaoPdAMartins/md-dmri (Nilsson et al., 2018b). The presented *in vivo* data may be directly requested from the authors.

## Acknowledgments

The authors thank Scott Hoge for his assistance with the MRI measurements. J.P.A.M. and D.T. were financially supported by the Swedish Foundation for Strategic Research (AM13-0090, ITM17-0267) and the Swedish Research Council (2014–3910, 2018-03697). C.M.W.T. is supported by a Rubicon grant (680-50-1527) from the Netherlands Organisation for Scientific Research (NWO). F.S. and C.-F.W. are both supported by a National Institutes of Health grant (P41EB015902). D.K.J. is supported by a Wellcome Investigator Award (096646/Z/11/Z) and a Wellcome Strategic Award (104943/Z/14/Z).

## Author Contributions

D.K.J., C.-F.W., and D.T. conceived the project. J.P.A.M., C.M.W.T., and F.S. designed the acquisition protocol. F.S. and C-F.W. acquired the data. The nonparametric Monte-Carlo algorithm was designed by J.P.A.M. and D.T, and the data analysis was performed by J.P.A.M., C.M.W.T. and D.T.. J.P.A.M., and D.T. wrote the manuscript, and all authors read and reviewed the manuscript.

## Competing interests

D.T. owns shares in and J.P.A.M. is partially employed by the private company Random Walk Imaging AB (Lund, Sweden), which holds patents related to the described method. All other authors declare no competing interests.

**A** Pulse sequence for 5D $R_2$-**D** encoding

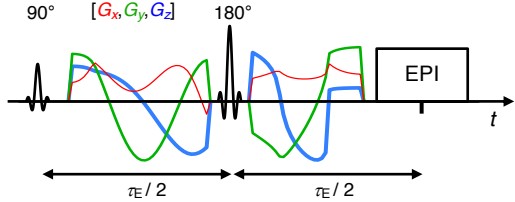

**B** Gradient waveforms for $b$-tensor encoding

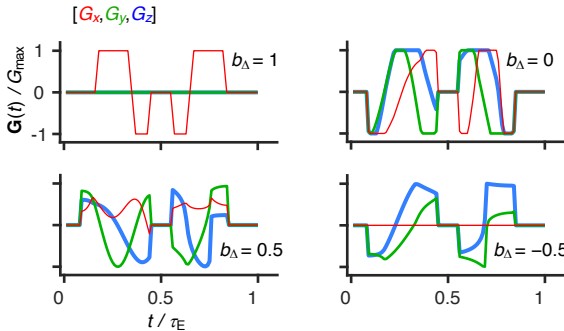

**Figure 1 Acquisition protocol for 5D relaxation–diffusion MRI.** (A) Pulse sequence for acquiring images encoded for relaxation and diffusion in a 5D space defined by the echo time $\tau_E$, and $b$-tensor trace $b$, normalized anisotropy $b_\Delta$, and orientation $(\Theta, \Phi)$. An EPI image readout block acquires the spin-echo produced by slice-selective 90° and 180° radio-frequency pulses. The 180° pulse is encased by a pair of gradient waveforms allowing for diffusion encoding according to principles from multidimensional solid-state NMR (Topgaard, 2017) (red, green, and blue lines). The signal is encoded for the transverse relaxation rate $R_2$ by varying the value of $\tau_E$. (B) Numerically optimized gradient waveforms (Sjölund et al., 2015) yielding four distinct $b$-tensor shapes ($b_\Delta = -0.5$, 0.0, 0.5, and 1) (Eriksson et al., 2015).

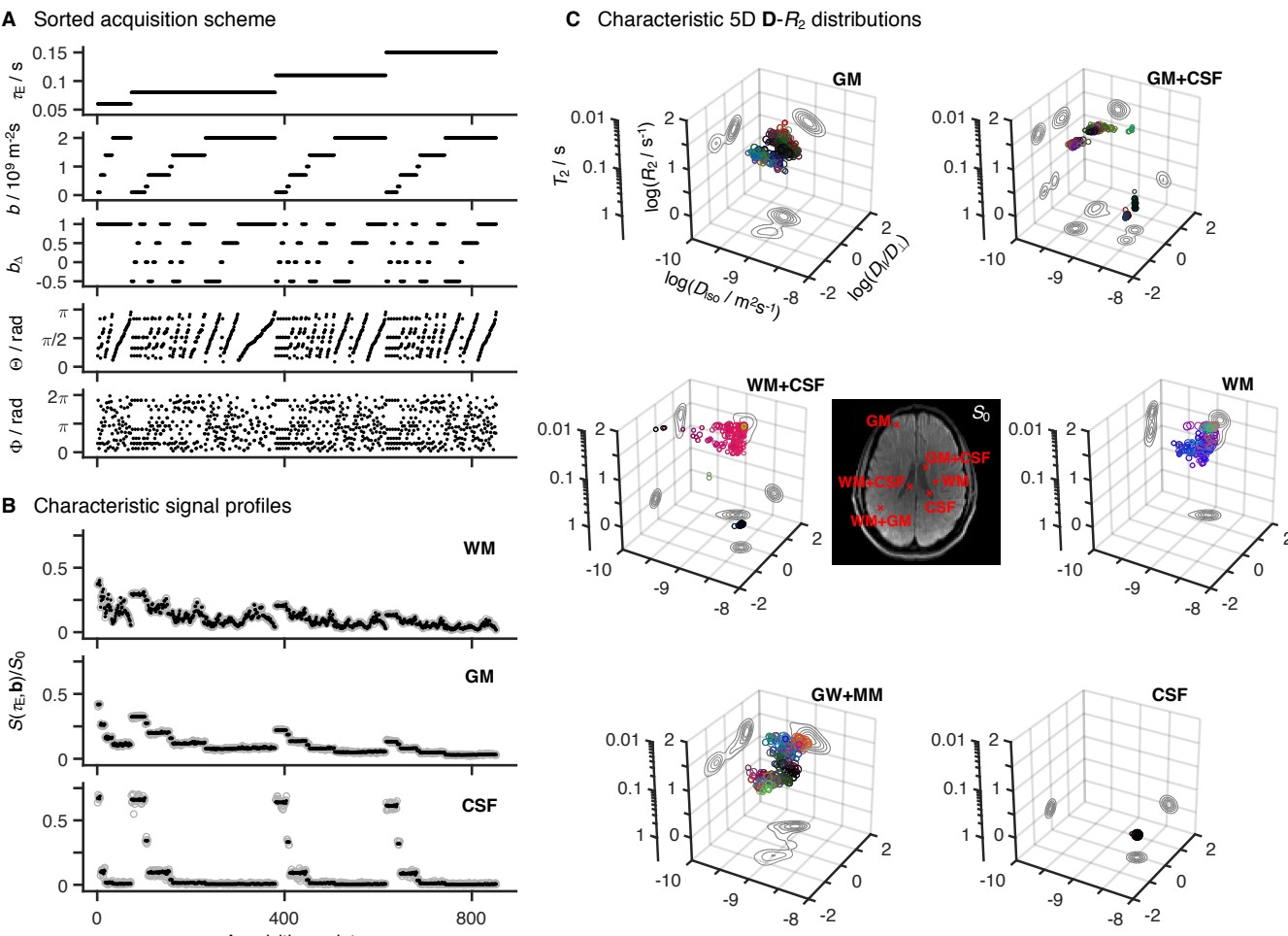

**Figure 2 Representative 5D relaxation-diffusion encoded signals $S(\tau_E,\mathbf{b})$ and distributions $P(R_2,\mathbf{D})$ for selected voxels in a living human brain.** (A) Acquisition scheme showing $\tau_E$, $b$, $b_\Delta$, $\Theta$, and $\Phi$ as a function of acquisition point. (B) Experimental (gray circles) and fitted (black points) $S(\tau_E,\mathbf{b})$ signals from three representative voxels containing white matter (WM), gray matter (GM), and cerebrospinal fluid (CSF). The presented signal data was acquired according to the scheme shown in panel A and is drawn with the same horizontal axis. (C) Nonparametric $R_2$-$\mathbf{D}$ distributions obtained for both pure (WM, GM, CSF) and mixed (WM+GM, WM+CSF, GM+CSF) voxels. The discrete distributions are reported as scatter plots in a 3D space of the logarithms of the transverse relaxation rate $R_2$, isotropic diffusivity $D_{iso}$, and axial-radial diffusivity ratio $D_\parallel/D_\perp$. An auxiliary relaxation time $T_2$ scale was included along the $\log(R_2)$ axis to aid the inspection of the $P(R_2,\mathbf{D})$ plots. The diffusion tensor orientation $(\theta,\phi)$ is color-coded as $[R,G,B] = [\cos\phi\sin\theta, \sin\phi\sin\theta, \cos\theta] \cdot |D_\parallel - D_\perp|/\max(D_\parallel, D_\perp)$ and the circle area is proportional to the statistical weight of the corresponding component. The contour lines on the sides of the plots represent projections of the 5D $P(R_2,\mathbf{D})$ distribution onto the respective 2D planes. Panels (B) and (C) display the signals $S(\tau_E,\mathbf{b})$ and corresponding $P(R_2,\mathbf{D})$, respectively, for the same WM, GM, and CSF voxels.

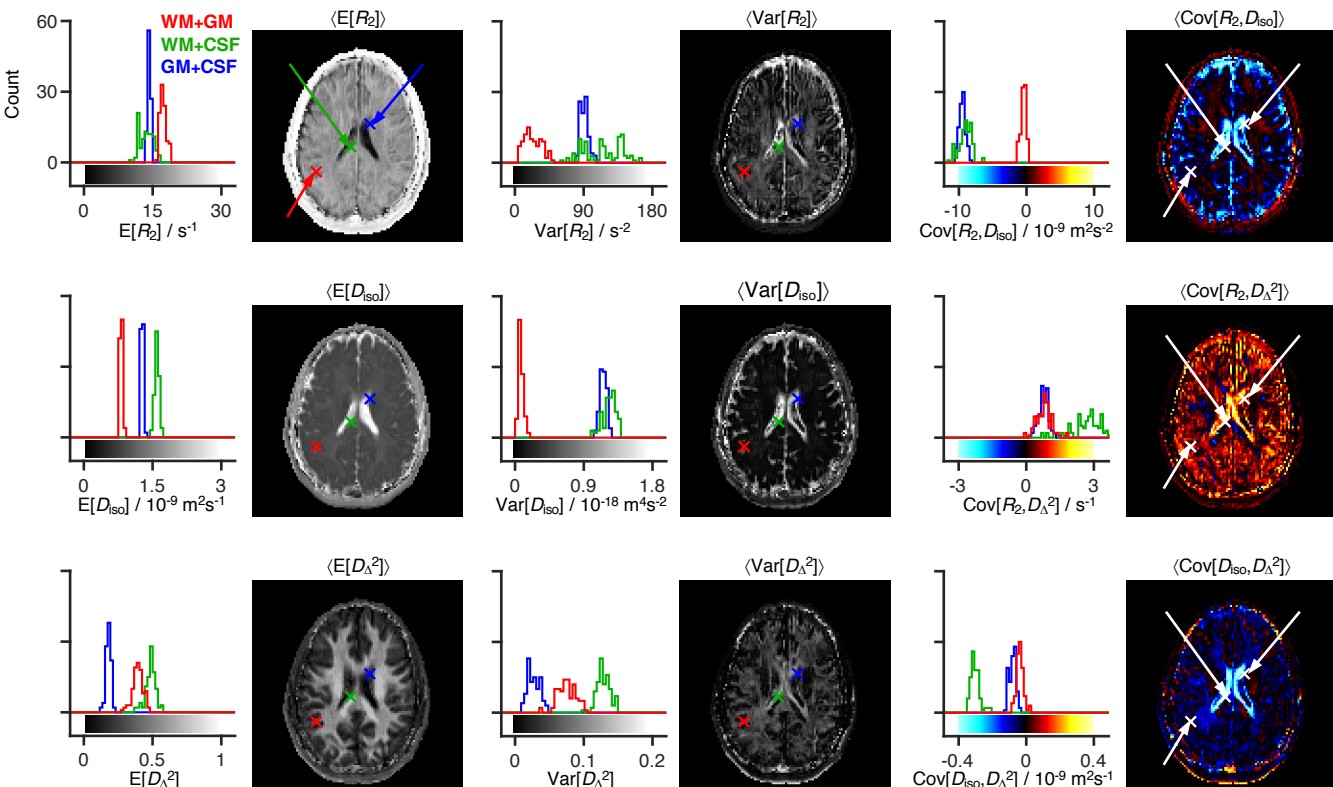

**Figure 3 Statistical measures derived from the relaxation-diffusion distributions.** The ensemble of 96 distinct $P(R_2,\mathbf{D})$ solutions was used to calculate means $E[x]$, variances $Var[x]$ and covariances $Cov[x,y]$ of all combinations of transverse relaxation rate $R_2$, isotropic diffusivity $D_{iso}$, and squared anisotropy $D_\Delta^2$. The statistical measures were all derived from the entire $R_2$-$\mathbf{D}$ distribution space on a voxel-by-voxel basis. Histograms are used to represent the parameter sets calculated for three voxels containing binary mixtures of white matter

WM, grey matter GM, and cerebrospinal fluid CSF. Each histogram comprises 96 estimates of a single statistical measure. The averages of statistical measures, $\langle E[x]\rangle$, $\langle Var[x]\rangle$ and $\langle Cov[x,y]\rangle$, are displayed as parameter maps whose color scales are given by the bars along the abscissas of the histograms. The crosses and arrows identify the heterogeneous voxels analyzed in the histograms; notice that the signaled points correspond to the average (as measured by the median) of the ensembles of plausible solutions shown in the histograms.

**Table 1** $R_2$-D limits of the 'Big', 'Thin', and 'Thick' bins

| | limits | $D_{iso}$ | | $D_{\parallel}/D_{\perp}$ | | $R_2$ | | $T_2$ | |
|---|---|---|---|---|---|---|---|---|---|
| | | $\log(x/\mathrm{m^2s^{-1}})$ | $x/10^{-9}\,\mathrm{m^2s^{-1}}$ | $\log(x)$ | $x$ | $\log(x/\mathrm{s^{-1}})$ | $x/\mathrm{s^{-1}}$ | $\log(x/\mathrm{s})$ | $x/\mathrm{s}$ |
| Big | Max | −8 | 10 | 3.5 | $3 \cdot 10^3$ | 2 | 100 | 0.5 | 3.3 |
| | Min | −8.7 | 2 | −3.5 | $3 \cdot 10^{-4}$ | −0.5 | 0.3 | −2 | 0.01 |
| Thin | Max | −8.7 | 2 | 3.5 | $3 \cdot 10^3$ | 2 | 100 | 0.5 | 3.3 |
| | Min | −10 | 0.1 | 0.6 | 4 | −0.5 | 0.3 | −2 | 0.01 |
| Thick | Max | −8.7 | 2 | 0.6 | 4 | 2 | 100 | 0.5 | 3.3 |
| | Min | −10 | 0.1 | −3.5 | $3 \cdot 10^{-4}$ | −0.5 | 0.3 | −2 | 0.01 |

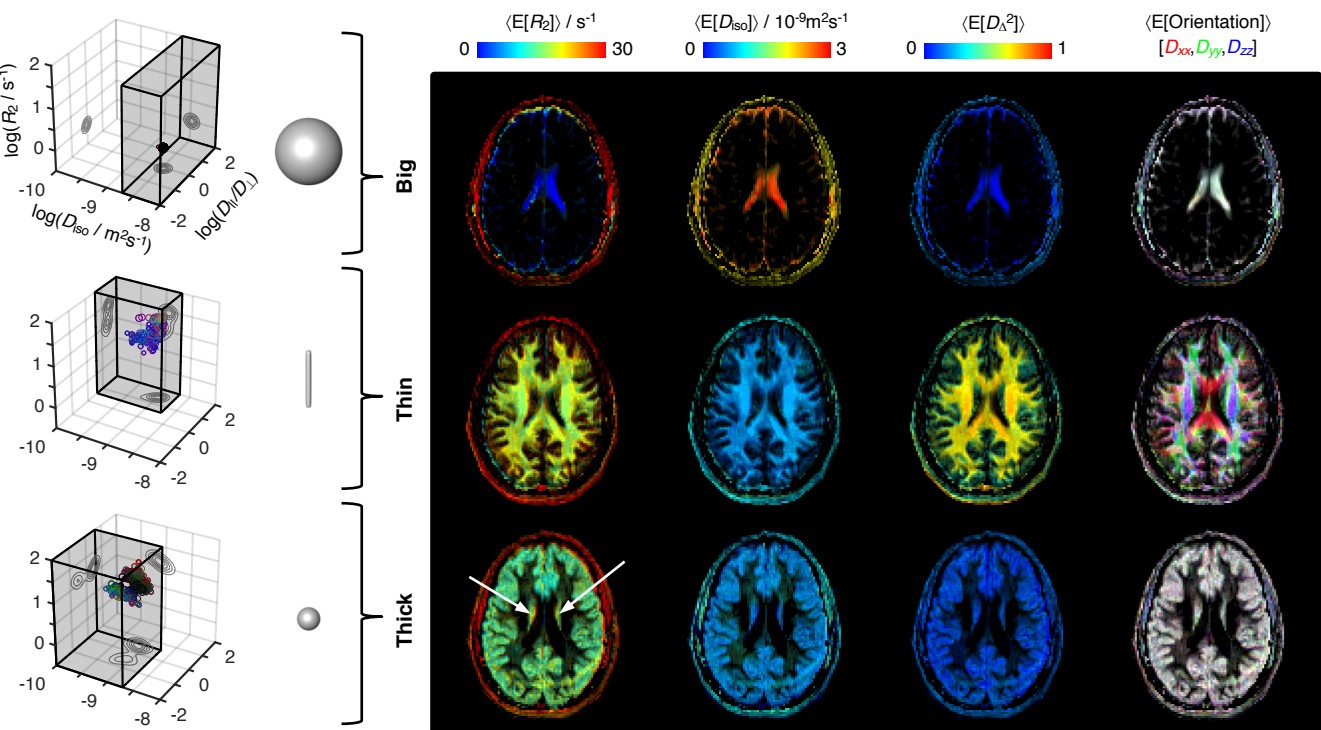

**Figure 4 Parameter maps with bin-resolved means of the relaxation-diffusion distributions.** (A) Division of the $R_2$-**D** distribution space into different bins. The distribution space was separated into three bins (gray volumes) named 'Big', 'Thin', and 'Thick' that loosely capture the diffusion features of cerebrospinal fluid CSF, white matter WM, and gray matter GM, respectively. The 3D scatter plots display the nonparametric $R_2$-**D** distributions corresponding to the CSF (top), WM (middle), and GM (bottom) voxels selected in **Figure 2**. Superquadratic tensor glyphs are used to illustrate the representative **D** captured by each bin. (B) Parameter maps of average per-bin means (color) of transverse relaxation rate $\langle E[R_2]\rangle$, isotropic diffusivity $\langle E[D_{iso}]\rangle$, squared anisotropy $\langle E[D_\Delta^2]\rangle$, and diffusion tensor orientation $\langle E[Orientation]\rangle$. The orientation maps (column 4) are color-coded as $[R,G,B] = [D_{xx}, D_{yy}, D_{zz}]/\max(D_{xx}, D_{yy}, D_{zz})$, where $D_{ii}$ are the diagonal elements of laboratory-framed average diffusion tensors estimated from the various distribution bins. Brightness indicates the signal fractions corresponding to the 'Big' (row 1), 'Thin' (row 2), and 'Thick' (row 3) bins. The white arrows identify deep gray matter structures.

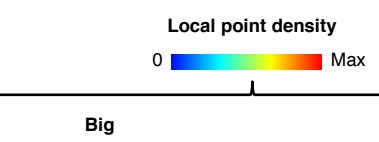

Figure 5 Uncertainty estimation of the statistical measures derived from the relaxation-diffusion distributions. 3D density (color) scatter plots show the relationship between average initial signal intensity $\langle S_0 \rangle$, the average of mean values derived from the $R_2$-**D** distributions $\langle E[x] \rangle$, and their corresponding uncertainties $\sigma[E[x]]$. For display purposes, signal intensity values were normalized to the maximum recorded $\langle S_0 \rangle$, $\max(\langle S_0 \rangle)$. The contour lines on the side planes show 2D projections of the point density function defining the distribution of data points. The average mean values of transverse relaxation rate $\langle E[R_2] \rangle$ (row 1), isotropic diffusivity $\langle E[D_{iso}] \rangle$ (row 2), and squared anisotropy $\langle E[D_\Delta^2] \rangle$ (row 3) were computed from all voxels whose $\langle S_0 \rangle$ was greater than 5% of $\max(\langle S_0 \rangle)$. The resulting dataset comprises 55327 voxels spread throughout all slices of the acquired 3D volume. The uncertainties of $\langle E[R_2] \rangle$, $\langle E[D_{iso}] \rangle$, and $\langle E[D_\Delta^2] \rangle$ correspond to the median absolute deviation between measures extracted from 96 independent solutions of Equation (2): $\sigma[E[R_2]]$, $\sigma[E[D_{iso}]]$, and $\sigma[E[D_\Delta^2]]$, respectively. All displayed data was derived from both the entire $R_2$-**D** space (column 1), and the 'Big' (column 2), 'Thin' (column 3), and 'Thick' (column 4) bins defined in **Figure 4A**.

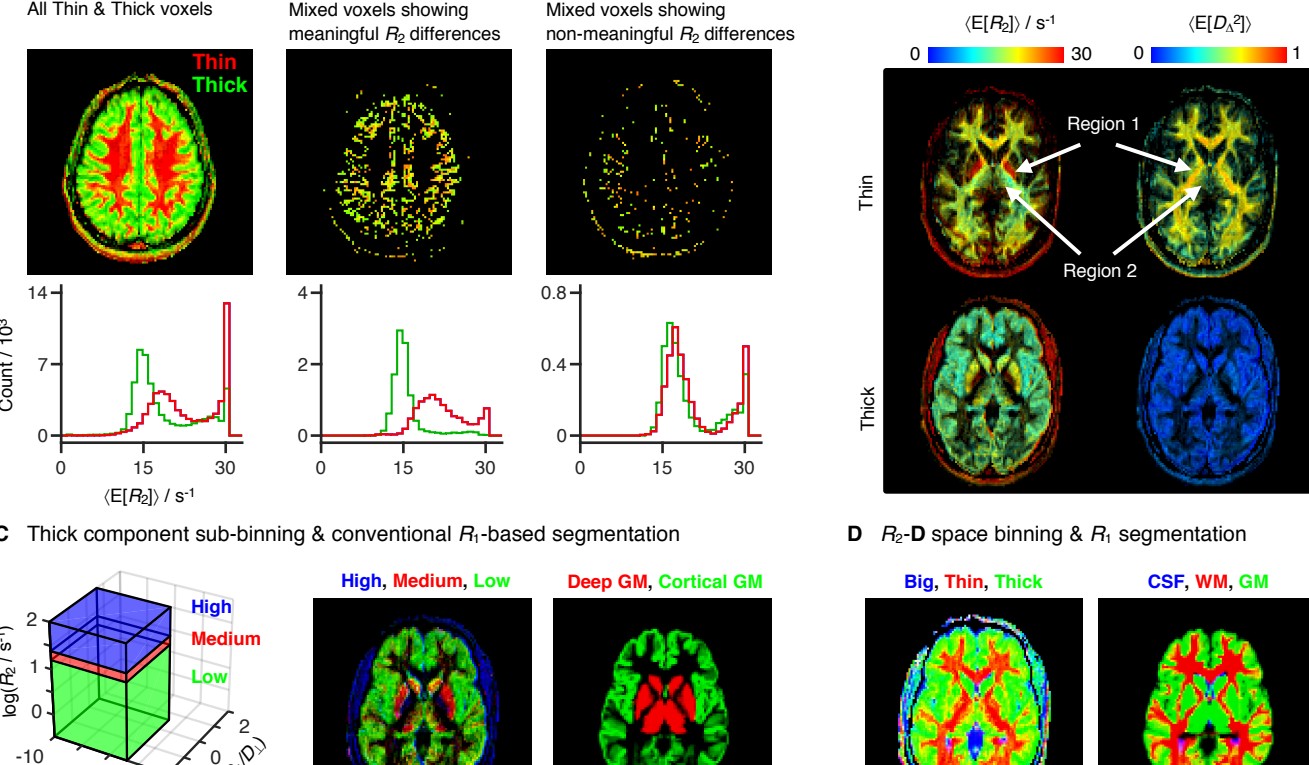

**Figure 6 Per-bin relaxation properties and tissue composition.** (A) Transverse relaxation properties specific to each of the 'Thin' (red) and 'Thick' (green) bins defined in **Figure 4A**. The color-coded composite images (top) and histograms (bottom) display the fractional populations and average mean transverse relaxation values $\langle E[R_2]\rangle$ of the two bins. The first column displays all of the 'Thin' and 'Thick'
voxels, while the two other columns focus on 'Thin'+'Thick' mixtures wherein the bin-specific $\langle E[R_2]\rangle$ values exhibit either significant (second column) or non-significant (third column) differences. (B) Bin-resolved signal fractions (brightness) and average per-bin means (color) of $R_2$, and squared anisotropy $D_\Delta^2$. Regions 1 and 2 identify microstructural properties singled-out in the Results section. (C) Subdivision of the 'Thick' bin into three different $R_2$ sub-spaces. The contributions from different sub-bins are compared with a high-resolution $R_1$-weighted image segmented into four different tissues: white matter WM, cortical gray matter GM, deep GM, and
cerebrospinal fluid CSF. Additive color maps display the spatial distribution of sub-bin fractions (from low to high $R_2$: green, red, blue), and of cortical (green) and deep (red) GM. (D) Color-coded composite images showing the contributions of different bins (red=Thin, green=Thick, blue=Big) and conventional $R_1$-based segmentation labels (red=WM, green=cortical+deep GM, blue=CSF).

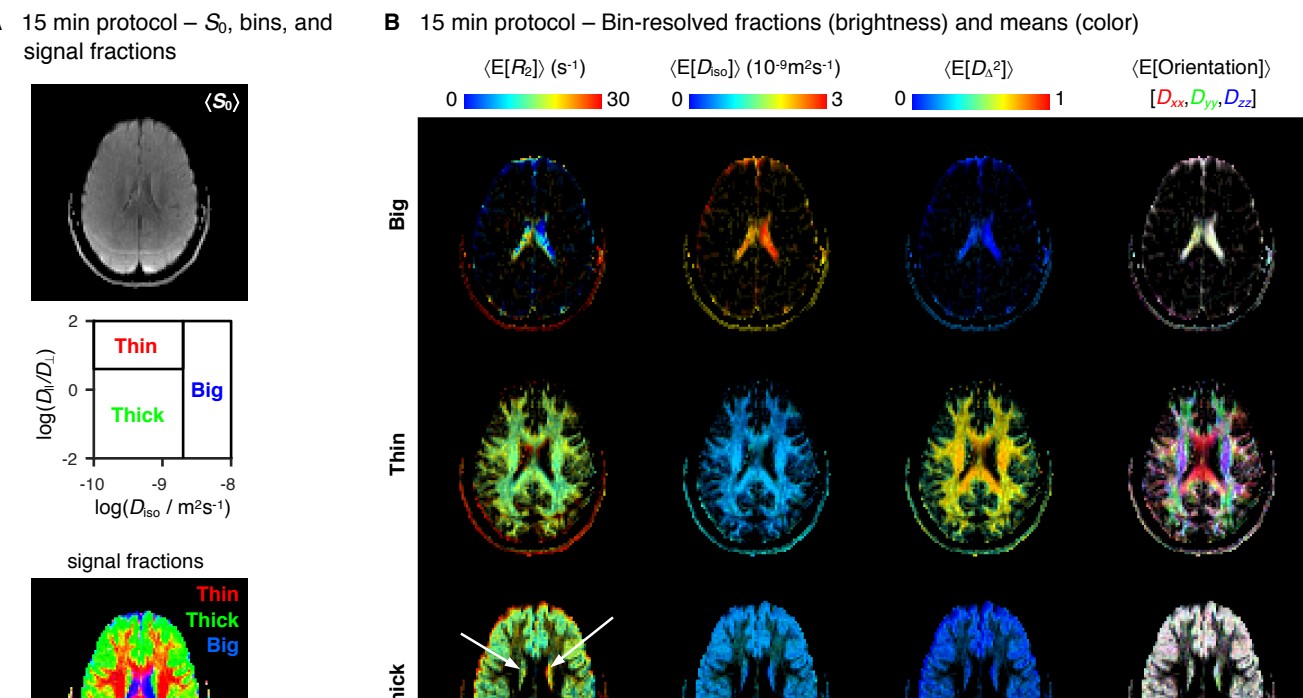

**A** 15 min protocol – $S_0$, bins, and signal fractions

$\langle S_0 \rangle$

$\log(D_\parallel/D_\perp)$ vs $\log(D_{iso}$ / m²s⁻¹)

Thin
Big
Thick

signal fractions

Thin
Thick
Big

**B** 15 min protocol – Bin-resolved fractions (brightness) and means (color)

$\langle E[R_2] \rangle$ (s⁻¹)   0 – 30
$\langle E[D_{iso}] \rangle$ (10⁻⁹m²s⁻¹)   0 – 3
$\langle E[D_\Delta^2] \rangle$   0 – 1
$\langle E[Orientation] \rangle$   [$D_{xx}$, $D_{yy}$, $D_{zz}$]

Big
Thin
Thick

**Figure 7 15 min protocol – Bin-resolved signal contributions and mean parameter maps.** (A) Map of average initial signal intensity $\langle S_0 \rangle$ (top); subdivision of the diffusion space into the 'Big', 'Thin', and 'Thick' bins (middle); color-coded composite map of per-bin signal contributions (bottom). The colors in the bottom identify the fractions from different bins: [R,G,B] = [Thin,Thick,Big]. (B) Parameter maps of average per-bin means (color) of transverse relaxation rate $\langle E[R_2] \rangle$, isotropic diffusivity $\langle E[D_{iso}] \rangle$, squared anisotropy $\langle E[D_\Delta^2] \rangle$, and diffusion tensor orientation $\langle E[Orientation] \rangle$. The color and brightness of the various maps follows the same convention of **Figure 4B**.

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
