# Peer review of "Transferring principles of solid-state and Laplace NMR to the field of *in vivo* brain MRI"

_Magnetic Resonance, 2019_

## Referee Comment (RC1) · Anonymous Referee #1 · 19 Dec 2019

This manuscript describes a new approach to simultaneously measuring T2 and diffusion for characterization of microstructure in central nervous system tissue. An experimental protocol is set up to directly explore T2 relaxation and the diffusion tensor without model fitting. A unique aspect of this study is that diffusion in the voxel is interpreted as distributions of axially symmetric diffusion tensors with mean diffusivities and tensor anisotropy, Ddelta. What makes this work very different from more conventional diffusion studies is that the gradient trajectories are designed to refocus signal from voxels which have distinct diffusion tensor shapes. The measurement protocol is designed to extract 5 parameters for each voxel: T2, Dpar, Dper, theta, phi. The results are presented in 3 D plots of T2, MD, Dpar/Dper.

This manuscript is one of a series several recent publications by this group and applies

the technology to human brain for the first time. Both the measurement and the analysis are quite sophisticated- this is an impressive accomplishment and a challenging manuscript to review due to the large number of unconventional steps in the experimental design.

All in all, this reviewer finds this work both exciting and impressive. It scores very high marks for novelty. The most exciting single result for this reviewer was the observation that grey matter generally is represented by two tensors which have similar Diso's but different tensor shapes. This work shows results from a single healthy volunteer; hence I see this as a glimpse of what might be possible with this technology. Clearly much more work is required to establish what happens in normal and, later, what happens with different brain pathologies.

I have a few comments, presented here in the order they appear in the manuscript.

1) The voxel size required for this measurement is large at 2x2x6 mmˆ3 or 2.3x2.3x5 mmˆ. This is bad enough in normal brain, but it may hamper its capability to extract microstructural information from small brain lesions. Is there potential for improved resolution?

2) I didn't see how long it took to do the analysis. Minutes, hours or days? The description of the analysis approach around line 150-155 is quite brief; is it possible to describe how weighting of points is determined?

3) The results in figure 3 are focused on the description of the 3D plots for GM, WM and CSF. Can the authors speculate more on the anatomical interpretation of these plots, especially for GM. It's one thing to say that GM has prolate and oblate components, can this be interpreted in terms of brain cell structures(s)?

4) I appreciate that log plots are a useful way to display the 3D plots; however, it would be much more intuitive for this reviewer if the data were presented in numbers that readers are familiar with. Logarithmically scaled plots are good, but the numbers on

the axes are much easier to interpret if they are numbers we are familiar with.

5) To continue, most people think of T2 as a number in seconds. Log(R2) means little to me. It's reasonable to do the analysis and plot scales logarithmically, but I prefer to see the actual numbers in plots.

6) To continue again, the definition of the three bins: Big, Thin and Thick (around line 250) would be much easier to read if they were in a table and if the various limits were provided in units most MRI readers are familiar with.

7) The discussion on how GM, WM and CSF contributions can be separated in large voxels with partial volume effects in normal CNS tissue seems longer than necessary. Is there a solid justification for the value in separating GM and WM and CSF when, as mentioned in the manuscript, T1 weighting easily enables segmentation of these components? However, discussing the similarities and differences between these three tissues in the 3D plots is very interesting. But it seems unlikely that the T2,D approach could be better than T1 for GM/WM segmentation.

8) Near line 285, there is a discussion about R2 bins. The TE times used are 60, 80, 110, 150ms. The four R2 bins correspond to 10ms to 40ms, 40ms to 63ms, 63ms to 3.16s. There are at least a couple of concerns here: i) with the first TE at 60ms, signals from most of the lower T2 range are not measurable. ii) a more serious issue is the selection of 63 ms as a boundary point. T2 times for normal GM and WM at 3T hover around 60ms. This data could probably separate T2's at 60- 100ms from T2s over 1 second, but by using 63ms as a boundary, GM and WM T2s for most structures will be equally likely to be in one or the other bins. If this is not the case, then there may be bias problems with the T2 estimations.

9) An important recent frontier in the conventional diffusion field has been the separation of isotropic and anisotropic water environments (e.g. DBSI, NODDI, CHARM, for characterisation of edema and inflammation. In their discussion near line 315, can the authors speculate on potential advantages, or disadvantages, of using the tuned

gradient trajectory approach over conventional diffusion acquisition approaches for extracting the signal from spins that undergo isotropic diffusion?

10) In Figure 6A, there seems to be a 'pile-up' of signal with T2 = 30ms, which especially in the 'Thin' box. Is this artifact or something else?

In summary, I like this work very much and look forward to seeing more work from this team.

---

## Referee Comment (RC2) · Anonymous Referee #2 · 8 Feb 2020

Authors present multi-dimensional (this time 5D) NMR imaging, to resolve compartments with different relaxation and diffusion properties. This is a very promising and interesting work which tackles one of the main limitations of quantitative MRI, lack of specificity.

Per-voxel diffusion MR signal is a summary statistics of multiple compartments with different biophysical properties. The conventional approaches to decompose these compartments are multi-compartment modeling or data driven signal decomposition. Propose technique is highly valuable because it adds an important degree of freedom (gained from acquisition domain) that enables tissue specific sensitization, and therefore improved specificity.

The experiment is nicely done, well evaluated and the paper is well structured and well

written. It is in camera ready in current form.

I have below thoughts/questions:

- This work is presented as a clinically feasible acquisition, but a 45 minute acquisition is not clinical. It is described and supplemented that the work can be shorten, but all the evaluation are done and conclusion are made based on a 45 minute scan (especially given that you state that SNR boosting repletion is essential). Although 45 minute is less than a usual clinical MRI scan ($\sim$1 hour), but if one spend 45 min on DWI, there will be no room for clinically essential scans such as FLAIR, T1w, T2, T2*/SWI.

- I was wondering if you compared regions with anatomically known microstructural differences to see how your measures compare. E.g. CST vs a short U-fiber

- Any specific reason for 2x2x6 configuration? do you think isotropic acquisition affect your measurements?

- How long the parameter extraction with MC took? what was the hardware used?

- "The acquired images were not subjected to any additional corrections (e.g. denoising or motion correction) before data inversion." Do you mean you did not do motion correction at all (screaming face emoji)? This is a 45 minute scan, and regardless of how Zen your participant was, there would be motion. did you at least assess the amount of motion. In particular, I would expect that the motion correction of high b-value, spherical acquisitions would be challenging. Did you average repetitions without motion correction?

- It would be useful to know the number of unknown parameters

- The paper only mention PGSE and how old it is, it would be more fair if alternative sequences (e.g. OGSE) are compared against this approach

Few additional minor suggestions:

- "The structure of the brain is shaped by both disease and normal developments on a

wide range of length scales." This sounds a bit strange.

- "insufficient for direct observation of individual cells, chemical and cellular features", what features? Microstructural? This can be confused with functional features

- I couldn't understand why delta(D) can be negative and why the range is -0.5 to 1. Please elaborate so those new to this technique could better understand the sequence.

- it could help less familiar readers if you mention that the linear encoding (b_delta = 1) is the same as conventional PGSE
* * *

---

## Author Comment (AC1) · 16 Feb 2020

**Comments from Anonymous Referee #1**

Received and published: 19 December 2019

1) The voxel size required for this measurement is large at 2x2x6 mm3 or 2.3x2.3x5 mm2. This is bad enough in normal brain, but it may hamper its capability to extract microstructural information from small brain lesions. Is there potential for improved resolution?

**Response:** Nonparametric multi-exponential inversion approaches are known to be particularly sensitive to the experimental noise, with lower SNR levels leading to less accurate results. With this in mind, we opted for large voxels in order to achieve a high SNR. Another reason to select large slices (and consequently large voxel-size) was to ensure a good brain coverage within the selected acquisition times (45 and 15 mins for the 2x2x6 and 2.3x2.3x5 configurations, respectively). The rationale behind our choice of voxel-size is now explained in the manuscript. It is of course possible to increase spatial resolution at a direct cost of SNR, which in turn might affect the accuracy and precision of the nonparametric inversion. To optimise the trade-off between SNR and voxel-size or to explicitly derive the SNR limits of our framework is however beyond the scope of this contribution.

There is however potential to boost the SNR per unit time and attain an improved resolution without substantially affecting the performance of the nonparametric inversion. In terms of hardware, the use ultra-high field (7T and above, including FDA-approval of 7T), and scanners with field gradients up to 300 mT/m (Setsompop, *Neuroimage*, 2013), can boost SNR and reduce echo-time-per-unit-b-value, respectively. From the acquisition perspective, multi-band acquisition schemes can speed up overall acquisition times. Moreover, using a spiral read-out (Wilm, *Magn. Reson. Med.*, 2017) instead of a traditional rectilinear EPI may further reduce the echo time, boosting SNR which can then be traded for higher spatial resolution. Finally, from the analysis side, denoising and/or joint reconstruction approaches (Veraart, *Neuroimage*, 2016; Bazin, *Front. Neurosci.*, 2019; Wang, *J. Magn. Reson. Imag.*, 2019; Haldar, *Magn. Reson. Med.*, 2020) could further enhance the SNR, allowing for a higher resolution. The discussion on possible strategies for achieving an increased SNR and a subsequent improved resolution has been added to the manuscript.

Changes: Text in lines 147-148 and lines 429-441.

2) I didn't see how long it took to do the analysis. Minutes, hours or days? The description of the analysis approach around line 150-155 is quite brief; is it possible to describe how weighting of points is determined?

**Response:** Determining 96 bootstrap solutions with the Monte Carlo analysis procedure took approximately 72 hours on a standard 12 cores computer. This information has been included in section 2.3. The various ( $R_2$ ,**D**) points were selected through the stochastic procedure explained in section 2.3, and their respective weights were determined by solving Eq. 2 from the main text with a standard non-negative least squares algorithm (Lawson and Hanson, *Solving least square problems*, 1974). To clarify this point, we have slightly expanded the description of the inversion algorithm. **Changes:** Text in lines 158-173 and lines 187-189.

*3)* The results in figure 3 are focused on the description of the 3D plots for GM, WM and CSF. Can the authors speculate more on the anatomical interpretation of these plots, especially for GM. It's one thing to say that GM has prolate and oblate components, can this be interpreted in terms of brain cell structures(s)?

**Response:** The oblate and prolate components observed in the GM distributions are an artefact explained by the fact that prolate ( $D_{\Delta} > 0$ ) and oblate ( $D_{\Delta}

Changes: Text in sub-section 3.1 (lines 219-230).

4) I appreciate that log plots are a useful way to display the 3D plots; however, it would be much more intuitive for this reviewer if the data were presented in numbers that readers are familiar with. Logarithmically scaled plots are good, but the numbers on the axes are much easier to interpret if they are numbers we are familiar with.

**Response:** In this contribution, the  $R_2$ -**D** distributions are displayed in log(x) variables to avoid overloading the axes of the corresponding 3D plots with minor ticks and to maintain a consistency with our previous works (see de Almeida Martins, *Sci. Rep.*, 2018). To facilitate the inspection of the 3D scatter plots we added a complementary  $T_2$  scale alongside the initially chosen log( $R_2$ ) axis.

Changes: Changes in Figure 2B.

5) To continue, most people think of T2 as a number in seconds. Log(R2) means little to me. It's reasonable to do the analysis and plot scales logarithmically, but I prefer to see the actual numbers in plots.

**Response:** See reply to point 4.

6) To continue again, the definition of the three bins: Big, Thin and Thick (around line 250) would be much easier to read if they were in a table and if the various limits were provided in units most MRI readers are familiar with.

**Response:** The limits of the three bins are now defined in a table, using both logarithmic and standard variables.

Changes: Addition of Table 1.

7) The discussion on how GM, WM and CSF contributions can be separated in large voxels with partial volume effects in normal CNS tissue seems longer than necessary. Is there a solid justification for the value in separating GM and WM and CSF when, as mentioned in the manuscript, T1 weighting easily enables segmentation of these components? However, discussing the similarities and differences between these three tissues in the 3D plots is very

interesting. But it seems unlikely that the (T2,D) approach could be better than T1 for GM/WM segmentation.

**Response:** The aim of the section highlighted by the reviewer is not to present our  $R_2$ -**D** correlation protocol as an alternative to  $T_1$ -w segmentation approaches, but rather to demonstrate its ability to separate the contributions from sub-voxel environments with different relaxation and diffusion properties. This, of course, cannot be done with a  $T_1$ -weighted image alone. We consider that the successful resolution of partial volume effects in normal CNS tissue provides evidence that the proposed algorithm can indeed resolve microscopic tissue environments with significantly different MR properties. Moreover, we would like to point out that a "segmentation" based on the retrieved  $R_2$ -**D** distributions is completely independent from  $T_1$ -w segmentation approaches and does not require the use of anatomical atlases. Hence, the proposed correlation approach may be seen not as an alternative, but rather a complement to high-resolution  $T_1$ -w protocols. For example, the information retrieved through the 5D distributions can be used to map a dispersion of  $R_2$ -**D** properties within a given anatomical structure or tissue class.

Changes: Text in Discussion and Conclusions (lines 355-361).

8) Near line 285, there is a discussion about R2 bins. The TE times used are 60, 80, 110, 150ms. The four R2 bins correspond to 10ms to 40ms, 40ms to 63ms, 63ms to 3.16s. There are at least a couple of concerns here: i) with the first TE at 60ms, signals from most of the lower T2 range are not measurable. ii) a more serious issue is the selection of 63 ms as a boundary point. T2 times for normal GM and WM at 3T hover around 60ms. This data could probably separate T2's at 60- 100ms from T2s over 1 second, but by using 63ms as a boundary, GM and WM T2s for most structures will be equally likely to be in one or the other bins. If this is not the case, then there may be bias problems with the T2 estimations.

**Response:** We would like to note that the  $T_2$  of the retrieved components are all constrained to the 33–1000 ms region due to the limits imposed by the Monte Carlo algorithm; no  $P(R_2, \mathbf{D})$  solutions exist with  $T_2 < 33$  ms (or, equivalently,  $R_2 > 30 \text{ s}^{-1}$ ). The  $T_2 = 10$  ms bin border was merely defined to render an aesthetically pleasing plot (see Figure 6C). This is now stated in the manuscript. The separation of WM and GM  $T_2$ 's is facilitated by the additional diffusion tensor dimensions rather than  $T_2$  alone. The WM/GM  $T_2$  separation is then presented as an example where diffusion tensor correlations offer potentially useful information that is invisible in lower dimensional techniques (*e.g.* 1D relaxometry). To better drive this point, we modified the relevant text in the *Discussion and Conclusions* section.

**Changes:** Text in sub-section 3.3 (lines 311-314) and in *Discussion and Conclusions* (lines 383-388).

9) An important recent frontier in the conventional diffusion field has been the separation of isotropic and anisotropic water environments (e.g. DBSI, NODDI, CHARM, for characterisation of edema and inflammation. In their discussion near line 315, can the authors speculate on potential advantages, or disadvantages, of using the tuned gradient trajectory approach over conventional diffusion acquisition approaches for extracting the signal from spins that undergo isotropic diffusion?

**Response:** An important advantage of the presented protocol compared to model-based diffusion MRI methods such as DBSI or NODDI is the fact that our approach does not rely on *a priori* assumptions about the number or properties of the underlying sub-voxel components. Instead of devising a signal model with complex priors and constraints, (some of those that

were made in the aforementioned papers have since been shown to be invalid), we add complexity to the signal acquisition stage in order to access more specific information and alleviate the assumptions required for data analysis. To achieve this increased specificity, it is essential to acquire data with gradient trajectories yielding *b*-tensors of different "shapes" ( $b_{\Delta}$ ). For the specific problem of separating between anisotropic and isotropic diffusion components, the additional  $b_{\Delta}$  dimension is expected to be particularly useful whenever **D** orientation dispersion is present. The *Discussion and Conclusions* section of the manuscript was expanded to accommodate these points. The major disadvantage of our approach is the increased scan time that may be necessary to accommodate the additional  $b_{\Delta}$  data points. **Changes:** Text in *Discussion and Conclusions* (lines 370-381).

**10) In Figure 6A, there seems to be a 'pile-up' of signal with T2 = 30ms, which especially in the 'Thin' box. Is this artifact or something else?**

**Response:** As mentioned in the manuscript, the high- $R_2$  components in the 'Thin' bin correspond to non-masked skull contributions and the fast-relaxing fiber populations found in the pallidum. The 'pile-up' *per se* is explained by a previously reported multi-exponential analysis artefact (see Saab, *Magn. Reson. Med.*, 1999), where fast-relaxing contributions are observed to concentrate in the maximum allowed  $R_2$  (or, equivalently, minimum allowed  $T_2$ ). In this manuscript, the inversion procedure was constrained to a maximum  $R_2$  of 30 s-1 (or, equivalently, minimum  $T_2$  of 33 ms), which corresponds to the noted signal 'pile-up'. The artefact is now explicitly mentioned in the manuscript.

Changes: Text in sub-section 3.3 (lines 303-306).

**Comments from Anonymous Referee #2**

Received and published: 8 February 2020

(1) This work is presented as a clinically feasible acquisition, but a 45 minute acquisition is not clinical. It is described and supplemented that the work can be shorten, but all the evaluation are done and conclusion are made based on a 45 minute scan (especially given that you state that SNR boosting repletion is essential). Although 45 minute is less than a usual clinical MRI scan (~1 hour), but if one spend 45 min on DWI, there will be no room for clinically essential scans such as FLAIR, T1w, T2, T2\*/SWI.

**Response:** We do agree that a 45 minutes scan is not compatible with the vast majority of clinical applications. The stated clinical feasibility refers to the 15 min protocol discussed in the Supplemental Material. To clarify and justify this claim we rephrased the abstract and included 15 min data in the main manuscript. A new sub-section was created where the abbreviated acquisition protocol is briefly compared to the exhaustive protocol. The maps derived from the 15 min dataset were observed to provide essentially the same conclusions as their 45 min counterpart, despite the fact of yielding slightly noisier maps. It is our opinion that this observation demonstrates that our protocol has indeed potential for clinical translation. Moreover, no SNR boosting repetition was used in either the 45 min or the 15 min protocols. However, we also refer this reviewer to our response to Reviewer 1 (critique point 1). There, we highlighted ways in which the SNR per unit time could be boosted by hardware (7T is now FDA-approved), ultra-strong gradients (e.g. Connectom's 300 mT/m or Feinberg's 200mTm gradients combined with 7T), multi-band acquisition, spiral-readouts, denoising and joint reconstruction. In this case, we were responding to the guestion of how resolution might be boosted. However, the same solutions will help (for a fixed resolution) to reduce the total amount of data needed for the same resultant SNR, and can help to push down total acquisition time further.

Finally, perhaps 'clinical' is a term used broadly in the neuroimaging literature. While a 45 minute scan cannot be inserted into a standard imaging MRI battery used in busy radiological clinics, it can be used in clinic-research protocols, where patients are frequently enrolled in acquisition protocols totaling over 90 minutes in length. As such, we believe our approach can yield invaluable insights into brain disease and normal development 'as is' and discounting its utility in clinical research would risk throwing the baby out with the bathwater.

To this end, we explicitly mention the clinico-research potential of the 45 min protocol while clarifying that the long scan time impedes its use for clinical practice.

Changes: New sub-section 3.4 & new Figure 7.

**(2) I was wondering if you compared regions with anatomically known microstructural differences to see how your measures compare. E.g. CST vs a short U-fiber**

**Response:** A comparison between regions with known microstructural differences is indeed interesting and may provide important insights on the accuracy, precision, and resolution limits of our  $R_2$ -**D** correlation approach. However, given the already extensive length of our manuscript, we decided to postpone such comparison to a future work. Regarding the suggestion of comparing CST with other fiber populations, we plan on pursuing the suggested task using an isotropic voxel configuration which renders the presented protocol more compatible with fiber tracking algorithms. Such experimental design has been recently presented in the 2019 meeting of the International Society for Magnetic Resonance in Medicine (ISMRM), see de Almeida Martins, Proc. Intl. Soc. Mag. Reson. Med., 2019. **Changes:** None.

(3) Any specific reason for 2x2x6 configuration? do you think isotropic acquisition affect your measurements?

**Response:** The 2x2x6 configuration was chosen to yield detailed axial maps while assuring a good coverage with a reduced number of slices. The rationale for the choice of the 2x2x6 configuration is now explained in *Methods* section. An isotropic voxel configuration is not expected to affect our measurements in any way, and has already been used in a recent work where the presented  $R_2$ -D correlation protocol is used to map and characterize fiber-tissues in heterogeneous voxels (see de Almeida Martins, Proc. Intl. Soc. Mag. Reson. Med., 2019). Please also see our detailed response to Reviewer 1 (comment 1). **Changes:** Text in the sub-section 2.2 (lines146-148)

**(4) How long the parameter extraction with MC took? what was the hardware used?**

**Response:** Data inversion with the presented Monte Carlo algorithm took approximately 72 hours on a standard 12 cores computer. The inversion time and the used hardware specs are now stated in section 2.3.

Changes: Text in lines 187-189.

(5) "The acquired images were not subjected to any additional corrections (e.g. denoising or motion correction) before data inversion." Do you mean you did not do motion correction at all (screaming face emoji)? This is a 45 minute scan, and regardless of how Zen your participant was, there would be motion. did you at least assess the amount of motion. In particular, I would expect that the motion correction of high b-value, spherical acquisitions would be challenging. Did you average repetitions without motion correction?

**Response:** The authors would like to reinforce that no averaging of repetitions was performed. Prior to the Monte Carlo analysis, we performed a quick visual comparison of the various images acquired in the 45 minutes protocol with no obvious motion artefacts being found. Despite these encouraging observations, we corrected the data for motion in ElastiX, using the extrapolated reference method detailed in Nilsson, *Plos One*, 2015. A quick 12 bootstrap inversion was conducted on both motion-corrected data and non-corrected data, and the resulting parameter maps were subsequently compared. As no substantial differences were found between the results from the corrected and non-corrected datasets, we opted to not use motion correction in our more comprehensive 96 bootstrap analysis. **Changes:** Text in sub-section 2.2 (lines 149-154).

**(6) It would be useful to know the number of unknown parameters**

**Response:** The number of unknown parameter changes at different steps of the inversion algorithm. The number of elements of the sought-for  $P(R_2, \mathbf{D})$  vector range from a few hundred at the initial iteration cycle where sets of  $(R_2, D_{\parallel}, D_{\perp}, \theta, \phi)$  points are randomly generated and then fitted to the data, to just 10 at the final step of the algorithm. However, we would like to point out that the exact number of unknowns at every step of the algorithm is difficult to define, as it will often depend on the number of non-zero weights found in a previous step. To better clarify the structure of the algorithm, we have expanded its description in the *Methods* section.

Changes: Text in sub-section 2.3 (lines 158-173).

(7) The paper only mention PGSE and how old it is, it would be more fair if alternative sequences (e.g. OGSE) are compared against this approach

**Response:** The multidimensional diffusion encoding strategy presented here has already been compared to alternative diffusion encoding schemes in recent reviews (see Topgaard, J Magn. Reson., 2017 or Topgaard, NMR methods for studying microscopic diffusion anisotropy in Diffusion NMR in Confined Systems, Ch. 7, 2016). For brevity, we decided to skip such comprehensive comparison and instead juxtapose our approach with classical PGSE sequences that are still the basis for the vast majority of *in vivo* diffusion MRI studies. Regarding the connection of this work with Oscillating Gradient Spin-Echo (OGSE) sequences, the authors do recognize that OGSE allows for a higher signal specificity due to its ability of selectively probing different structural length-scales. However, it is particularly challenging to achieve high *b*-values when using OGSE in a standard clinical scanner. Hence, it is difficult to probe the signal regime where anisotropy effects manifest with OGSEbased sequences. As the main advantage of our diffusion encoding strategy is its ability to probe and disentangle the effects of anisotropy and orientation dispersion, we feel that a comparison with OGSE wouldn't add any meaningful insight to the manuscript. However, it should be noted that OGSE methods should be mentioned and discussed if the current protocol is extended to include gradient waveforms with a wide array of spectral profiles and a kernel that explicitly accounts for the effects of time-dependent diffusion (see discussion in lines 379-391).

Changes: None.

Few additional minor suggestions:

(1) "The structure of the brain is shaped by both disease and normal developments on a wide range of length scales." This sounds a bit strange.
Response: We have rephrased the pointed sentence.
Changes: Text in line 24.

(2) "insufficient for direct observation of individual cells, chemical and cellular features", what features? Microstructural? This can be confused with functional features
 Response: The term "cellular features" can indeed generate confusion. We have thus rephrased it to "microstructural features".
 Changes: Text in line 30.

(3) I couldn't understand why delta(D) can be negative and why the range is -0.5 to 1. Please elaborate so those new to this technique could better understand the sequence.

**Response:** The  $D_{\Delta}$  metric is used to quantify the anisotropy of **D** and its definition draws inspiration from the field of solid-state NMR (see Topgaard, *J. Magn. Reson.*, 2017), where a similar metric is used to parameterize the anisotropy of the chemical-shift tensor. Negative values of  $D_{\Delta}$  can be found when water molecules are constrained to an oblate pore structure where the radial diffusivity,  $D_{\perp}$ , is higher than axial diffusivity,  $D_{\parallel}$  (*e.g.:* water diffusion in lamellar liquid crystals). The range of  $D_{\Delta}$  follows immediately from its definition:  $D_{\Delta} = (D_{\parallel} - D_{\perp})/(D_{\parallel} + 2D_{\perp})$ . In the limit  $D_{\parallel} \rightarrow 0$  (diffusion within a 2D plane structure),  $D_{\Delta} = -0.5$ . Similarly, for  $D_{\perp} \rightarrow 0$  (diffusion within a stick),  $D_{\Delta} = 1$ .

(4) it could help less familiar readers if you mention that the linear encoding (b\_delta = 1) is the same as conventional PGSE

**Response:** In the context of this work, the Stejskal-Tanner designator refers to conventional PGSE schemes. To clarify this point we have rephrased some text in the introduction and explicitly mention that data acquisition with a conventional PGSE corresponds to diffusion encoding with  $b_{\Delta} = 1$ .

Changes: Text in lines 54 and 93.